# A Ca$^{2+}$-regulated deAMPylation switch in human and bacterial FIC proteins

Simon Veyron [1], Giulia Oliva[2,3], Monica Rolando[2], Carmen Buchrieser[2], Gérald Peyroche[1] & Jacqueline Cherfils [1]

FIC proteins regulate molecular processes from bacteria to humans by catalyzing post-translational modifications (PTM), the most frequent being the addition of AMP or AMPylation. In many AMPylating FIC proteins, a structurally conserved glutamate represses AMPylation and, in mammalian FICD, also supports deAMPylation of BiP/GRP78, a key chaperone of the unfolded protein response. Currently, a direct signal regulating these FIC proteins has not been identified. Here, we use X-ray crystallography and in vitro PTM assays to address this question. We discover that *Enterococcus faecalis* FIC (EfFIC) catalyzes both AMPylation and deAMPylation and that the glutamate implements a multi-position metal switch whereby Mg$^{2+}$ and Ca$^{2+}$ control AMPylation and deAMPylation differentially without a conformational change. Remarkably, Ca$^{2+}$ concentration also tunes deAMPylation of BiP by human FICD. Our results suggest that the conserved glutamate is a signature of AMPylation/deAMPylation FIC bifunctionality and identify metal ions as diffusible signals that regulate such FIC proteins directly.

[1] CNRS and Ecole normale supérieure Paris-Saclay, Laboratoire de Biologie et Pharmacologie Appliquée, 61 Avenue du Président Wilson, 94235 Cachan CEDEX, France. [2] Institut Pasteur and CNRS UMR 3525, Biologie des Bactéries Intracellulaires, 25-28 Rue du Dr Roux, 75015 Paris, France. [3] Sorbonne Université, Collège doctoral, 75005 Paris, France. Correspondence and requests for materials should be addressed to J.C. (email: jacqueline.cherfils@ens-paris-saclay.fr)

In less than a decade, FIC (filamentation induced by cAMP) proteins have emerged as a large family of enzymes controlling the activity of target proteins by posttranslationally modifying them with phosphate-containing compounds (reviewed in refs. [1–4]). These proteins are characterized by the presence of a conserved FIC domain, which carries out the posttranslational modification (PTM) of a Tyr, Ser, or Thr residue in a target protein[5–12]. The most frequent PTM reaction catalyzed by FIC enzymes is the addition of AMP using ATP as a cofactor, coined AMPylation or adenylylation. This PTM activity was originally discovered in toxins from bacterial intracellular pathogens[5] and later identified in the toxin component of bacterial toxin/antitoxin (TA) modules (e.g., Bartonella VbhT/VbhA[8]) and in various other bacterial FIC proteins of unknown functions, including single-domain (e.g., Neisseria FIC[8]) and larger (e.g., Clostridium FIC[12]) FIC proteins. Metazoans possess a single FIC protein, FICD/HYPE (FICD hereafter), which controls the reversible AMPylation of BiP/GRP78, an Hsp70 chaperone localized in the endoplasmic reticulum (ER)[13–15]. BiP is a key component of the unfolded protein response (UPR), a major pathway whereby cells respond to ER stress (reviewed in ref. [16]). AMPylation of BiP reduces its affinity for unfolded protein clients[17] and inversely correlates with the burden of unfolded proteins[15]. In a recent twist, FICD was shown to act bifunctionally in both AMPylation and deAMPylation of BiP and to carry out deAMPylation as its primary enzymatic activity in vitro[18,19].

All PTM reactions catalyzed by FIC proteins use a motif of conserved sequence for catalysis, the FIC motif, which carries an invariant histidine that is critical for nucleophilic attack of the cofactor, and an acidic residue (aspartate or glutamate) that binds an $Mg^{2+}$ ion to stabilize the negative charges of the cofactor phosphates at the transition state (reviewed in refs. [1–4]). A large group of FIC proteins also features a structurally conserved glutamate that represses AMPylation by protruding into the catalytic site from either N-terminal elements, as in human FICD[11], Clostridum difficileFIC[12], or Shewanella oneidensis FIC[20] or from a C-terminal α-helix as in single-domain FIC proteins from Neisseria meningitidis[8], where repression of AMPylation was first described, and Helicobacter pylori (PDB 2F6S). Crystal structures showed that this glutamate prevents ATP from binding in a catalytically competent conformation[8,12], while its mutation into glycine creates space for the γ-phosphate of ATP[8] and increases AMPylation activities in vitro and in cells (reviewed in ref. [21]). These observations led to propose that this conserved glutamate represses AMPylation by impairing the utilization of ATP as a donor for AMP, hence that it must be displaced to allow productive binding of ATP[8]. In N. meningitidis FIC (NmFIC), upregulation of the AMPylation activity was shown to occur upon changes in the toxin concentration[22]. In this scheme, NmFIC is an inactive tetramer at high concentration, which is further

stabilized by ATP, while conversion into a monomer upon decrease of enzyme concentration leads to upregulation of its AMPylation activity, which is further reinforced by subsequent autoAMPylations[22]. Whether a similar mechanism applies to other glutamate-bearing FIC proteins has not been investigated. Remarkably, in bifunctional human FICD, the autoinhibitory glutamate is critical for deAMPylation of the BiP chaperone[18], a reaction whereby FICD contributes to the activation of BiP under ER stress conditions. Currently, a signal that represses the deAMPylation activity of FICD under normal ER conditions has not been identified. Together, these intriguing observations raise the question of whether displacement of the autoinhibitory glutamate is the sole mode of regulation of glutamate-containing AMPylating FIC proteins, calling for the identification of diffusible signals able to lift autoinhibition of AMPylation and, in the case of FICD, to activate deAMPylation.

In this study, we address this question by combining structural analysis and PTM assays of a single-domain FIC protein from Enterococcus faecalis (EfFIC), which carries an autoinhibitory glutamate in C-terminus, and of human FICD. Enteroccoci are commensals of the gastrointestinal tract that become pathogenic outside of the gut and cause nosocomial infections through acquisition and transmission of antibiotic resistance[23,24]. We discover that EfFIC has both AMPylation and deAMPylation activities borne by the same active site. Furthermore, the conserved glutamate implements a $Ca^{2+}$-controlled metal switch that tunes the balance between these activities without conformational change. Finally, we show that $Ca^{2+}$ is an inhibitor of deAMPylation of BiP by human FICD. Our findings predict that the presence of a glutamate is a signature for AMPylation/deAMPylation bifunctionality in FIC proteins from bacteria to human, and they identify metals as diffusible signals that can directly modulate the activity of glutamate-bearing FIC proteins by binding into the active site. Importantly, they suggest that FICD has features of an enzymatic $Ca^{2+}$ sensor, which may respond to the drop in $Ca^{2+}$ concentration, a hallmark of ER stress, with implications in therapeutic strategies against diseases that involve the UPR.

## Results

**EfFIC is an AMPylator.** Enterococcus faecalis FIC belongs to class III FIC proteins, which are comprised of a single FIC domain and carry an autoinhibitory glutamate in their C-terminal α-helix. We determined crystal structures of unbound, phosphate-bound, AMP-bound, and ATPγS-bound wild-type EfFIC ($EfFIC^{WT}$) and of unbound and sulfate-bound EfFIC carrying a mutation of the catalytic histidine into an alanine ($EfFIC^{H111A}$) (Table 1 and Supplementary Table 1). These structures were obtained in 5 different space groups, yielding 32 independent copies of the EfFIC monomer with various environments in the

**Table 1 Summary of the crystal structures of EfFIC determined in this study**

| EfFIC | Space group | Cell parameters a, b, c (Å) | Ligand | Resolution | PDB |
|---|---|---|---|---|---|
| WT | $P4_12_12$ | 65.13, 65.13, 248.06 | $PO_4^{2-}$ | 2.29 | 6ER8 |
| WT | I222 | 121.54, 131.00, 136.94 | — | 2.40 | 5NV5 |
| WT | $P4_12_12$ | 64.98, 64.98, 246.24 | $AMP-Ca^{2+}$ | 2.35 | 6EP0 |
| WT | $P4_32_12$ | 125.35, 125.35, 362.8 | $ATPγS^a-Ca^{2+}$ | 2.15 | 6EP2 |
| WT | $P4_32_12$ | 87.84, 87.84, 364.94 | $ATPγS^a$ | 1.93 | 6EP5 |
| H111A | $P2_122_1$ | 76.67, 77.11, 103.15 | — | 2.60 | 5NWF |
| H111A | I222 | 121.93, 131.16, 136.71 | $SO_4^{2-}$ | 2.20 | 5NVQ |

All crystals have cell angles of α, β, and γ = 90°. Crystallographic statistics are given in Supplementary Table 1
EfFIC FIC protein from Enterococcus faecalis, WT wild type
[a]Only the ADP moiety is visible

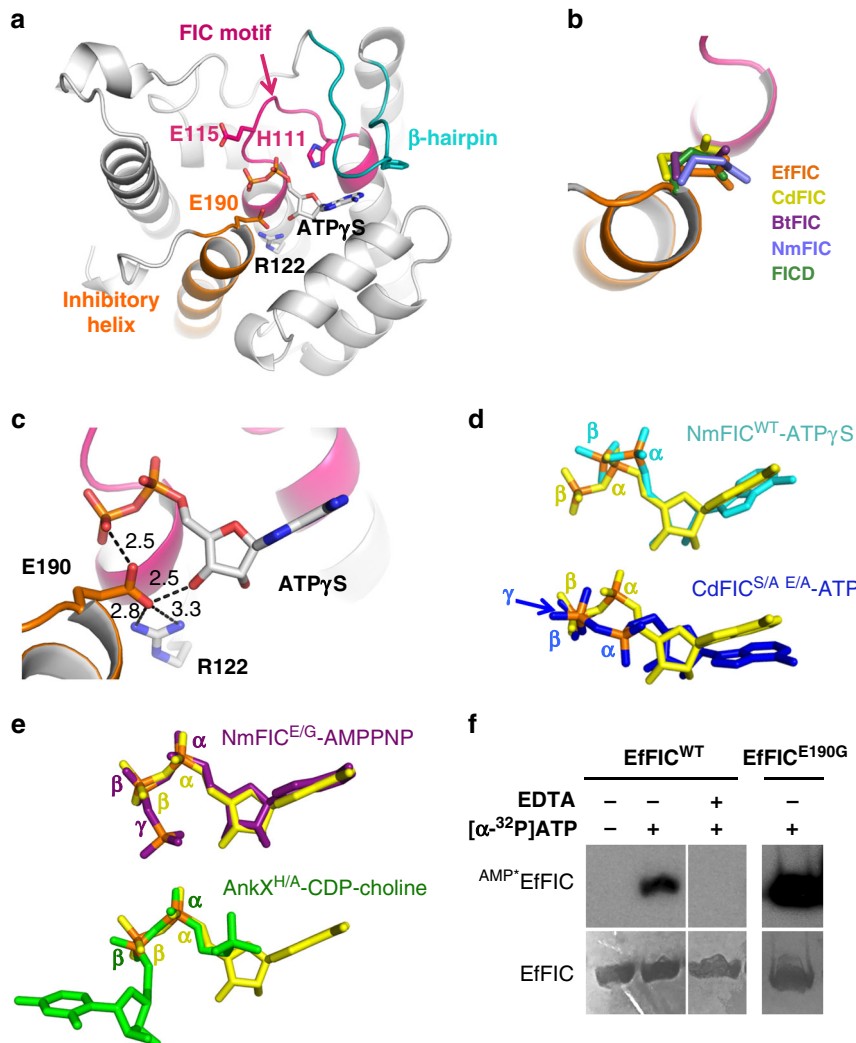

**Fig. 1** Structural basis for the AMPylation activity of FIC protein from *Enterococcus faecalis* (EfFIC). **a** Structure of the EfFIC monomer showing the FIC motif (pink), the C-terminal α-helix bearing the inhibitory glutamate (orange), and the β-hairpin predicted to bind protein substrates (cyan). The ADP moiety of ATPγS is shown in sticks. **b** The inhibitory glutamate from EfFIC[WT] (orange) is structurally equivalent to the glutamate found in the C-terminus of NmFIC (ref. [8], PDB 2G03) and in the N-terminus of *Bacteroides* BtFIC (PDB 3CUC), *Clostridium* CdFIC (ref. [12], PDB 4×2E) and human FICD/HYPE (ref. [11], PDB 4U04). Superpositions are done on the structurally highly conserved FIC motif. **c** Interactions of the inhibitory glutamate with the active site of ATPγS-bound EfFIC[WT]. Hydrogen bonds are depicted by dotted lines. **d** The positions of the α- and β-phosphates of ATPγS bound to EfFIC[WT] (yellow) diverge from those of ATPγS bound to NmFIC[WT] (cyan, ref. [8], PDB 3S6A) and of ATP bound to CdFIC (blue, ref. [12], S31A/E35A mutant, PDB 4×2D) in a non-canonical conformations. Note that only the ADP moiety of ATPγS is visible in the EfFIC[WT] and NmFIC[WT] crystal structures. **e** The α- and β-phosphates of ATPγS bound to EfFIC[WT] (yellow) superpose well to those of AMPPNP bound to NmFIC (purple, ref. [8], E186G mutant, PDB 3ZLM) and of CDP-choline bound to AnkX (green, ref. [9], H229A mutant, PDB 4BET) in a posttranslational modification-competent conformation. **f** AutoAMPylation of EfFIC[WT] and EfFIC[E190G]. The level of AMPylated proteins (indicated as AMP*EfFIC[WT]) was measured by autoradiography using radioactive [α-32P]-ATP in the presence of 100 nM Mg[2+]. The reaction was carried out for 1 h for EfFIC[WT] and 5 min for EfFIC[E190G]. The total amount of EfFIC[WT] measured by Coomassie staining in the same sample is shown

crystal. All EfFIC monomers resemble closely to each other and to structures of other class III FIC proteins (Fig. 1a). Notably, the C-terminal α-helix that bears the inhibitory glutamate (E190) shows no tendency for structural flexibility, even in molecules where it is free of crystalline contacts. E190 has the same conformation as in other glutamate-bearing FIC protein structures (Fig. 1b) and is stabilized by a salt bridge with Arg122 and, when present, by interactions with the nucleotide cofactor (Fig. 1c). Two crystal structures were obtained by co-crystallization with a non-hydrolyzable ATP analog (ATPγS), for which well-defined electron density was observed for the ADP moiety (Supplementary Fig. 1a). The positions of the α and β phosphates of ATPγS in these structures depart markedly from those seen in ATP bound

unproductively to wild-type NmFIC[8] or bound non-canonically to CdFIC[12] (Fig. 1d). In contrast, they superpose well to cofactors bound in a position competent for PTM transfer[8,9] (Fig. 1e). These observations prompted us to assess whether EfFIC is competent for AMPylation, using autoAMPylation as a convenient proxy in the absence of a known physiological target (reviewed in ref. [21]). Using [α-32P]-ATP and autoradiography to measure the formation of AMPylated EfFIC (denoted AMP*EfFIC[WT]), we observed that EfFIC[WT] has conspicuous autoAMPylation activity in the presence of Mg[2+] (Fig. 1f). Mutation of the inhibitory glutamate into glycine (E190G) increased AMPylation, indicating that the AMPylation efficiency of EfFIC[WT] is not optimal (Fig. 1f). We conclude from these

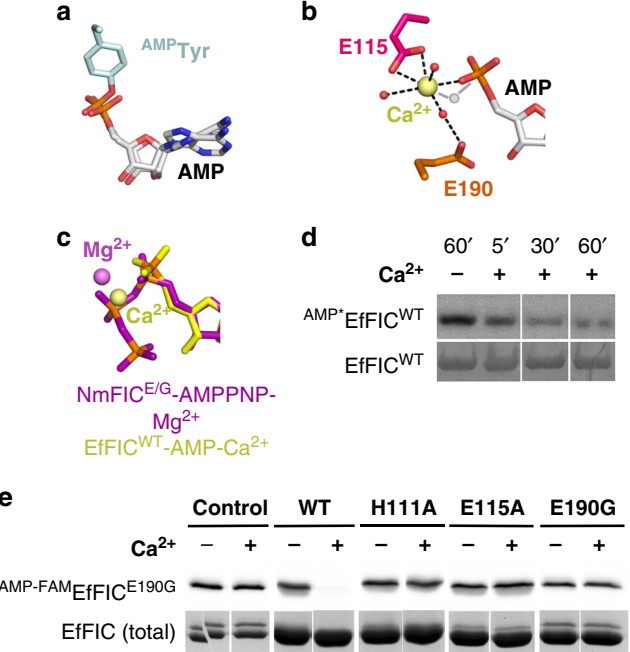

**Fig. 2** FIC protein from *Enterococcus faecalis* (EfFIC) is a deAMPylator in the presence of $Ca^{2+}$. **a** AMP bound to EfFIC$^{WT}$ superposes with the AMP moiety of AMPylated CDC42 in complex with the FIC protein IbpA (ref. [6], PDB 4ITR). Superposition was carried out based on the FIC motif. **b** Close-up view of $Ca^{2+}$ bound to EfFIC$^{WT}$-AMP. Interactions of $Ca^{2+}$ are indicated by dotted lines. Water molecules are shown as red spheres. The predicted position of the water molecule responsible for in-line nucleophilic attack is indicated in gray. Note that this water molecule completes the heptahedral coordination sphere of $Ca^{2+}$. **c** $Ca^{2+}$ bound to EfFIC$^{WT}$-AMP is shifted by 1.3 Å with respect to $Mg^{2+}$ bound to NmFIC$^{E/G}$-AMPPNP. The superposition is done on the FIC motif. **d** EfFIC$^{WT}$ has deAMPylation activity in the presence of $Ca^{2+}$. EfFIC$^{WT}$ was autoAMPylated with [α-$^{32}$P]-ATP in the presence 100 nM $Mg^{2+}$ for 1 h, then the sample was incubated for 60 min with EDTA alone (1 mM) (lane −) or 5, 30, and 60 min with EDTA (1 mM) and an excess of $Ca^{2+}$ (10 mM) (lanes +). AMPylation levels were analyzed by autoradiography (upper panel). The total amount of EfFIC$^{WT}$ in each sample was measured by Coomassie staining (lower panel). **e** Key residues of the AMPylation active site are required for deAMPylation. EfFIC$^{E190G}$ was left to auto-AMPylate for 1 h in the presence of the fluorescent ATP analog ATP-FAM and $Mg^{2+}$, then $^{AMP-FAM}$EfFIC$^{E190G}$ was purified to remove $Mg^{2+}$, PPi, and ATP-FAM. DeAMPylation was then triggered by addition of wild-type or mutant EfFIC in the presence or not of 1 mM $Ca^{2+}$. AMPylation levels (indicated as $^{AMP-FAM}$EfFIC$^{E190G}$) after 1 h incubation were analyzed by fluorescence (upper panel). The total amount of EfFIC proteins (indicated as EfFIC$^{total}$) in each sample was measured by Coomassie staining (lower panel)

experiments that wild-type EfFIC has canonical features of an AMPylating FIC enzyme and that the inhibitory glutamate mitigates the efficiency of AMPylation.

## EfFIC is a deAMPylator in the presence of $Ca^{2+}$.
To gain further insight into the enzymatic properties of EfFIC, we solved the crystal structure of EfFIC$^{WT}$ bound to AMP, used as a surrogate of the product of the AMPylation reaction (Table 1 and Supplementary Table 1). AMP superposes exactly to the AMP moiety of AMPylated CDC42 in complex with the FIC2 domain of the IbpA toxin[6], supporting its subsequent analysis as a mimic of the AMPylated product (Fig. 2a). Electron-rich density was observed next to AMP in the active site (Supplementary Fig. 1b), ascribed to a calcium ion present in the crystallization solution to the

exclusion of all other metal ions. Accordingly, the ion coordination sphere has a pentagonal bipyramid configuration with bond lengths in the 2.1–2.9 Å range, which is incompatible with bond lengths and fixed octahedral geometry of $Mg^{2+}$ but is often encountered for $Ca^{2+}$ (reviewed in ref. [25]). $Ca^{2+}$ binds to six oxygen atoms, including a phosphate oxygen of AMP, the carboxylate oxygen of the acidic residue in the FIC motif (Glu115), and three water molecules, one of which is also bound to the inhibitory glutamate (Glu190), with the seventh ligand missing (Fig. 2b). The position of $Ca^{2+}$ in the EfFIC$^{WT}$-AMP structure is shifted with respect to that of $Mg^{2+}$ observed in other FIC protein structures in complex with ATP (Fig. 2c), raising the intriguing possibility that it may support an alternative biochemical activity. Inspired by the recent observation that animal FICD proteins have deAMPylation enzymatic activity[18,19], we analyzed whether EfFIC could have deAMPylation activity in the presence of $Ca^{2+}$. Remarkably, addition of $Ca^{2+}$ to EfFIC$^{WT}$ that had been first autoAMPylated by incubation with $Mg^{2+}$ and [α-$^{32}$P]-ATP resulted in a decrease of AMPylated EfFIC$^{WT}$ (Fig. 2d), revealing a potent deAMPylation activity.

In the above set-up, the AMPylation and deAMPylation activities are potentially acting concurrently. To characterize the deAMPylation reaction selectively, the hyperactive EfFIC$^{E190G}$ mutant was autoAMPylated in the presence of $Mg^{2+}$, purified to remove ATP, PPi, and $Mg^{2+}$ such that no AMPylation reaction remains possible, and then its deAMPylation was triggered by addition of EfFIC$^{WT}$ or EfFIC mutants in the presence of $Ca^{2+}$. The level of AMPylated EfFIC was quantified by fluorescence using ATP-FAM, an ATP analog fluorescently labeled on the adenine base (denoted $^{AMP-FAM}$EfFIC). Robust deAMPylation of $^{AMP-FAM}$EfFIC$^{E190G}$ was observed upon addition of EfFIC$^{WT}$ and $Ca^{2+}$ (Fig. 2e, WT panel). No deAMPylation was observed in the absence of EfFIC$^{WT}$ (Fig. 2e, control panel), indicating that the deAMPylation reaction occurs in trans. We used this deAMPylation set-up to identify residues critical for deAMPylation catalysis. Mutation of the catalytic histidine (H111A), of the metal-binding acidic residue in the FIC motif (E115A), or of the inhibitory glutamate (E190G) completely impaired deAMPylation of $^{AMP-FAM}$EfFIC$^{E190G}$ (Fig. 2e, mutant panels). We conclude from these experiments that EfFIC is a bifunctional enzyme and that the AMPylation and deAMPylation activities are borne by the same catalytic site. To better reflect the dual role of the inhibitory glutamate in AMPylation and deAMPylation, we refer to this residue as the conserved glutamate in the rest of the study.

## AMPylation and deAMPylation are regulated by metals.
The above results raise the issue of the nature of signals acting on the bifunctional active site of EfFIC to regulate AMPylation/deAMPylation alternation. Previous work showed that AMPylation of *Escherichia coli* DNA gyrase by NmFIC, which shares 56% sequence identity with EfFIC, was highly sensitive to the toxin concentration, with a sharp drop of activity >250 μM arising from the conversion of NmFIC from a monomer to a tetramer[22]. EfFIC forms equivalent dimers in all crystal forms (Supplementary Fig. 2), but the arrangement of these dimers in higher-order assemblies varies between crystals (Supplementary Table 1). Depending on the crystal forms, EfFIC does not form assemblies larger than the dimers (Supplementary Fig. 2a) or forms tetramers related to the NmFIC tetramer (Supplementary Fig. 2b) or forms hexamers comprised of trimers of dimers (Supplementary Fig. 2c), making it difficult to establish the functional oligomeric state based on the crystallographic assemblies. Alternatively, we analyzed whether the deAMPylation activity of EfFIC$^{WT}$ would be sensitive to dilution, thus paralleling the mechanism described for NmFIC AMPylation. As shown in Supplementary Fig. 2d,

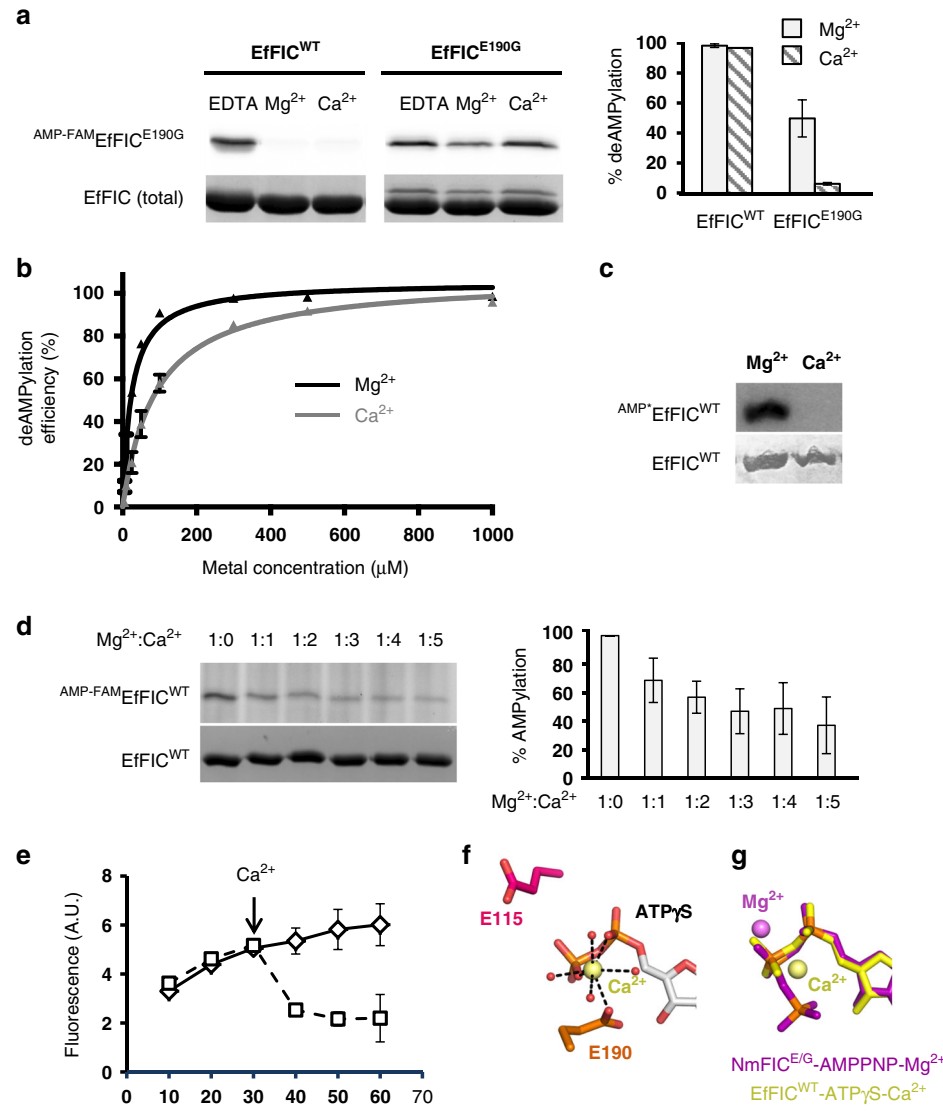

**Fig. 3** The $Mg^{2+}/Ca^{2+}$ ratio controls the balance between AMPylation and deAMPylation. **a** The conserved glutamate of FIC protein from *Enterococcus faecalis* (EfFIC) is required for deAMPylation in the presence of $Mg^{2+}$ and $Ca^{2+}$. AMP-FAMEfFIC$^{E190G}$ was incubated with 6 µM EfFIC$^{WT}$ and either 0.5 mM EDTA or 3 mM $Mg^{2+}$ or $Ca^{2+}$. DeAMPylation was quantified by fluorescence (right panel). **b** Determination of the apparent dissociation constants of $Ca^{2+}$ and $Mg^{2+}$ for EfFIC$^{WT}$. EfFIC$^{WT}$ (4 µM) was incubated with 5 µM ATP-FAM, then deAMPylation was carried out in the presence of increasing metal concentrations. $K_D$ are 25 µM ($R^2 = 0.98$) for $Mg^{2+}$ and 90 µM ($R^2 = 0.99$) for $Ca^{2+}$. **c** $Ca^{2+}$ inhibits EfFIC$^{WT}$ autoAMPylation. AMPylation was carried out for 1 h at 1 mM $Ca^{2+}$ or $Mg^{2+}$ using radiolabeled ATP and measured by autoradiography. **d** The net AMPylation efficiency of EfFIC$^{WT}$ is controlled by the $Mg^{2+}/Ca^{2+}$ ratio. EfFIC$^{WT}$ was incubated for 1 h with ATP-FAM at fixed $Mg^{2+}$ concentration (1 mM) and increasing $Ca^{2+}$concentrations. The steady-state AMPylation efficiency was quantified by fluorescence and expressed as the percentage of the maximal AMPylation level obtained with $Mg^{2+}$ alone (right panel). All measurements have $p$ values <0.05 with respect to the experiment containing $Mg^{2+}$ alone. The total amount of EfFIC$^{WT}$ is shown (Coomassie staining). **e** The EfFIC activity switch is controlled by $Ca^{2+}$. EfFIC$^{WT}$ (8 µM) was incubated with 1 mM $Mg^{2+}$, then the reaction was started by addition of 10 µM ATP-FAM. The level of ATP-FAMEfFIC$^{WT}$ was quantified at the indicated time points using fluorescence. At $t = 30$ min, 5 mM $Ca^{2+}$ (squares) or buffer (diamonds) was added and the reactions were carried out for another 30 min. **f** Close-up view of $Ca^{2+}$ bound to EfFIC$^{WT}$-ATPγS. The pentagonal bipyramidal coordination sphere of $Ca^{2+}$ is indicated by dotted lines. Water molecules are in red. Note that only the ADP moiety of ATPγS is visible in the electron density. **g** $Ca^{2+}$ bound to EfFIC$^{WT}$-ATPγS is shifted with respect to $Mg^{2+}$ bound to AMPylation-competent NmFIC$^{E/G}$-AMPPNP (PDF 3ZLM). All measurements were done in duplicate except in **e** (triplicate). Error bars correspond to standard deviations. Source data are provided as a Source Data file

deAMPylation of purified AMP-FAMEfFIC$^{E190G}$ by EfFIC$^{WT}$ increased as the concentration of EfFIC$^{WT}$ increased from 1 to 2000 nM. Thus deAMPylation is not adversely affected by the concentration of the toxin. Alternatively, we reasoned that the distinct electrochemical properties of $Ca^{2+}$ and $Mg^{2+}$ (reviewed in ref. [25]) may allow these metals to support AMPylation and deAMPylation differentially. First, we compared the ability of

$Mg^{2+}$ and $Ca^{2+}$ to support the deAMPylation activity of EfFIC$^{WT}$. As shown in Fig. 3a (left panel), both metals supported potent deAMPylation. Mutation of the conserved glutamate eliminated deAMPylation in the presence of $Ca^{2+}$, as observed above (Fig. 1e), while partial deAMPylation was retained in the presence of $Mg^{2+}$ (Fig. 3a, right panel). We took advantage that $Mg^{2+}$ and $Ca^{2+}$ both support deAMPylation to determine their

apparent dissociation constants ($K_D$) using titration curves, yielding a $K_D$ of 25 and 90 μM for $Mg^{2+}$ and $Ca^{2+}$, respectively (Fig. 3b). These values are in the same range for both metals and in the same range as dissociation constants reported for bacterial $Mg^{2+}$ transporters (reviewed in ref. [26]), consistent with a biological function for both metals in deAMPylation. In striking contrast, while $Mg^{2+}$ supported potent AMPylation, AMPylation was completely eliminated in the presence of saturating concentration of $Ca^{2+}$ (Fig. 3c). Furthermore, $Ca^{2+}$ also reduced the net AMPylation efficiency in the presence of $Mg^{2+}$ in a manner that depended on the $Mg^{2+}/Ca^{2+}$ ratio (Fig. 3d). The ability of $Ca^{2+}$ to revert AMPylation supported by $Mg^{2+}$ was further confirmed in a kinetics assay, showing that addition of $Ca^{2+}$ suffices to switch EfFIC from the AMPylation regime to the deAMPylation regime (Fig. 3e). To understand how $Ca^{2+}$ inhibits AMPylation, we determined the crystal structure of $EfFIC^{WT}$-$ATPγS$-$Ca^{2+}$ (Supplementary Table 1 and Supplementary Fig. 1c). In this structure, $Ca^{2+}$ is heptacoordinated to the α- and β-phosphates of ATPγS (of which again only the ADP moiety is visible), to the inhibitory glutamate, and to four water molecules (Fig. 3f). Strikingly, $Ca^{2+}$ is shifted by 3.2 Å from $Mg^{2+}$ bound to NmFIC and ATP[8] (Fig. 3g). In this position, $Ca^{2+}$ conflicts with the binding of $Mg^{2+}$ to ATP, and it cannot interact with the acidic residue from the FIC motif (E115), which is critical for AMPylation. These observations explain why $Ca^{2+}$ inhibits AMPylation and how it competes with $Mg^{2+}$.

Together, these observations depict a three-position metal switch differentially operated by $Mg^{2+}$ and $Ca^{2+}$, including an AMPylation-competent position where $Mg^{2+}$ binds in the presence of the ATP cofactor, a deAMPylation-competent position where $Ca^{2+}$ and $Mg^{2+}$ bind in the presence of an AMPylated substrate, and an AMPylation inhibitory position where $Ca^{2+}$ binds in the presence of ATP. We conclude from these experiments that competition between $Mg^{2+}$ and $Ca^{2+}$ controls the net balance between AMPylation and deAMPylation and that this regulatory metal switch is implemented by differential usage of the inhibitory glutamate and the acidic residue in the FIC motif for binding metals.

**$Ca^{2+}$ tunes deAMPylation of the BiP chaperone by human FICD.** DeAMPylation of the BiP chaperone has been recently identified as the primary activity of human FICD in vitro[18]. FICD features a glutamate structurally equivalent to the inhibitory glutamate in EfFIC (see Fig. 1b)[11], which is critical for deAMPylation[18]. We analyzed whether, as observed in EfFIC, $Mg^{2+}$ and $Ca^{2+}$ metals could also affect FICD activity, using fluorescent ATP-FAM to monitor BiP AMPylation. No measurable AMPylation of BiP by wild-type FICD ($FICD^{WT}$) could be observed, regardless of whether the metal is $Mg^{2+}$ or $Ca^{2+}$ (Fig. 4a), consistent with previous observations using autoradiography that the AMPylation activity of FICD is intrinsically repressed[8,14]. We note that $FICD^{WT}$ itself showed autoAMPylation, as also observed for Drosophila $FICD^{WT}$[14], which occurs in the presence of both metals (Fig. 4a). Alternatively, we used $FICD^{E234G}$, in which the inhibitory glutamate is mutated into glycine, to produce and purify AMPylated BiP ($^{AMP\text{-}FAM}BIP$). Remarkably, while $^{AMP\text{-}FAM}BIP$ was efficiently deAMPylated by $FICD^{WT}$ in the presence of $Mg^{2+}$, no deAMPylation was measured in the presence of $Ca^{2+}$ (Fig. 4b). To determine whether FICD does not bind $Ca^{2+}$ or is unable to use it for deAMPylation, we carried out an $Mg^{2+}/Ca^{2+}$ competition experiment in which $FICD^{WT}$ and purified $^{AMP\text{-}FAM}BiP$ were incubated at increasing $Ca^{2+}$ concentrations and a fixed $Mg^{2+}$ concentration. As shown in Fig. 4c, deAMPylation efficiency decreased as the $Ca^{2+}/Mg^{2+}$ ratio increased, suggesting that $Ca^{2+}$ inhibits deAMPylation by

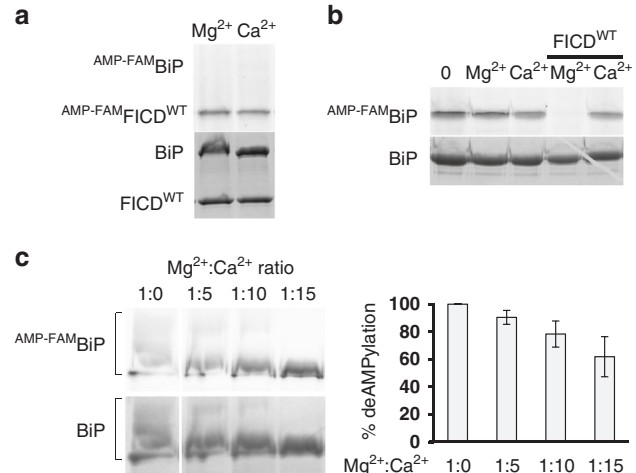

**Fig. 4** DeAMPylation of BiP by FICD requires $Mg^{2+}$ and is inhibited by $Ca^{2+}$. **a.** $FICD^{WT}$ does not AMPylate the BiP chaperone. Reactions were carried out for 1 h in the presence of 1 mM $Mg^{2+}$ or 1 mM $Ca^{2+}$. The level of AMPylated proteins (indicated as $^{AMP\text{-}FAM}FICD^{WT}$ and $^{AMP\text{-}FAM}BiP$) was measured by fluorescence. The total amount of proteins measured by Coomassie staining in the same sample is shown. **b** $Ca^{2+}$ inhibits deAMPylation of BiP by FICD/HYPE$^{WT}$. BiP was first AMPylated for 1 h by the hyperactive $FICD^{E234G}$ mutant in the presence of 100 μM $Mg^{2+}$, then $^{AMP\text{-}FAM}BiP$ was purified to remove $Mg^{2+}$, PPi, and ATP-FAM. DeAMPylation of $^{AMP\text{-}FAM}BiP$ was then triggered by addition of wild-type or mutant FICD in the presence of 1 mM $Mg^{2+}$ or 1 mM $Ca^{2+}$. $^{AMP\text{-}FAM}BiP$ levels were determined by measuring the fluorescence intensity of each band. The total amount of BiP measured by Coomassie staining in each sample is shown. **c** The net deAMPylation efficiency of $FICD^{WT}$ is tuned by the $Mg^{2+}/Ca^{2+}$ ratio. DeAMPylation was carried out as in **b** using a fixed $Mg^{2+}$ concentration (200 μM) and 0, 1, 2 or 3 mM $Ca^{2+}$. AMPylation levels were measured by fluorescence, normalized to the fluorescence intensity of FICD, and expressed as the percentage of maximal deAMPylation level obtained in the absence of $Ca^{2+}$. All data have $p$ values <0.05 with respect to the control in the absence of $Ca^{2+}$. The total amount of BiP measured by Coomassie staining in each sample is shown. All experiments were done in duplicate. Error bars correspond to standard deviations. Source data are provided as a Source Data file

competing with $Mg^{2+}$. We conclude from these experiments that $Ca^{2+}$ binds to FICD in an inhibitory manner, which allows it to tune its deAMPylation efficiency toward the BiP chaperone.

## Discussion

In this study, we sought after a diffusible signal able to regulate directly the large group of AMPylating FIC proteins in which the AMPylation activity is repressed by a conserved glutamate within the active site. Combining crystallography and PTM assays, we first show that bacterial EfFIC is a bifunctional enzyme that encodes AMPylating and deAMPylating activities and that both reactions use the same active site. Next, we discover that the balance between these opposing activities is controlled by a metal switch, in which each reaction is differentially supported and inhibited by $Mg^{2+}$ and $Ca^{2+}$ in a manner that the $Mg^{2+}/Ca^{2+}$ ratio determines the net AMPylation efficiency. Furthermore, we identify the inhibitory glutamate and the acidic residue in the FIC motif as residues essential for the metal switch. Finally, we show that deAMPylation of the ER BiP chaperone by human FICD is also dependent on the $Ca^{2+}/Mg^{2+}$ ratio, in a manner that high $Ca^{2+}$ concentration inhibits deAMPylation.

The identification of a potent deAMPylation activity in a bacterial FIC protein (this study) and in human FICD[18,19], which depends on a structurally equivalent glutamate in these otherwise

**Fig. 5** Model of the Ca$^{2+}$-assisted deAMPylation catalytic mechanism. In this model, the regulatory glutamate (Glu190 in EfFIC (FIC protein from *Enterococcus faecalis*)) attracts a proton from a water molecule coordinating the metallic cation (step 1) to activate the nucleophilic attack of its oxygen on the phosphorus of the AMP moiety of the AMPylated substrate (step 2). The positive charge provided by Ca$^{2+}$ increases electrophilicity of the phosphorus and stabilizes the negative charge of the intermediate (steps 2 and 3). The intermediate harboring a pentavalent phosphorus then rearranges, leading to the breaking of the phosphor-ester bond, which is elicited by the capture of a proton provided by the catalytic histidine (step 4)

remotely related FIC proteins, leads us to propose that the conserved glutamate is a signature of the ability of FIC proteins to catalyze both AMPylation and deAMPylation. Our data allow to delineate the catalytic basis for this bifunctionality, in which all catalytic residues are shared by the AMPylation and deAMPylation reactions but have different roles in catalysis. In the AMPylation reaction, the invariant histidine in the FIC motif activates the acceptor hydroxyl of a target protein by attracting a proton, and the acidic residue (Asp or Glu) in the FIC motif binds a metal that stabilizes the phosphates of the ATP cofactor at the transition state (reviewed in refs. [2,3]). Based on the observations that the α- and β-phosphates of ATP bind with canonical positions to wild-type EfFIC and that AMPylation is potentiated by mutation of the conserved glutamate in various FIC proteins, we propose that the primary role of this glutamate in AMPylation is to mitigate the efficiency of this reaction in the presence of Mg$^{2+}$, rather than to fully repress it, possibly in order to match AMPylation and deAMPylation efficiencies. In the deAMPylation mechanism depicted in Fig. 5 (see also discussion in Supplementary Discussion and Supplementary Fig. 3), the conserved glutamate activates a water molecule for nucleophilic attack of the phosphorus of the AMP moiety of the AMPylated substrate, and the invariant histidine generates the free hydroxyl group in the protein residue by giving up a proton, as also proposed in ref. [18]. The nucleophilic water molecule is readily identified as the missing ligand needed to complete the heptahedral coordination sphere of Ca$^{2+}$ in the EfFIC-AMP-Ca$^{2+}$ structure, where it would be precisely positioned for in-line nucleophilic attack (Fig. 2b). It is likely that Ca$^{2+}$ also contributes to catalysis by stabilizing negative charges that develop at the transition state in the AMPylated substrate.

Importantly, our data identify a diffusible signal able to modulate the activity of glutamate-bearing FIC proteins. A remarkable feature in the above bifunctional mechanism is that both AMPylation and deAMPylation can be adversely regulated by a second metal that competes with the catalytic metal. In EfFIC, Ca$^{2+}$ binds to ATP in a manner that hinders binding of Mg$^{2+}$ to the canonical AMPylation metal-binding site, thereby inhibiting AMPylation. In a related scenario, Ca$^{2+}$ downregulates the deAMPylation activity of FICD by binding in an inhibitory manner where it competes with Mg$^{2+}$. This multi-position metal switch constitutes a new paradigm in bifunctional enzyme regulation, in which the relative concentrations of specific metals for the AMPylation and deAMPylation configurations tip the balance toward opposing activities within the same active site. Future studies are now needed to determine the bifunctionality spectrum of glutamate-bearing FIC proteins resulting from variations in metal specificities and affinities. In addition, observations by us and others[14] that FICD has distinct AMPylation and deAMPylation patterns toward itself and BiP suggest that the protein substrate influences the AMPylation/deAMPylation balance through still unknown mechanisms, which will have to be investigated. Another important question for future studies is how the metal switch mechanism combines with previously described mechanisms of regulation, including autoAMPylation and changes in oligomerization (reviewed in ref. [27]), such as those observed for the close homolog NmFIC[22]. These mechanisms may combine as multiple regulatory layers or they could operate under different physiological conditions.

Our findings predict that bacterial glutamate-containing FIC proteins are bifunctional enzymes able to switch between AMPylation and deAMPylation activities upon changes in metal homeostasis. An appealing hypothesis is thus that these FIC proteins function as TA modules in which the toxin and antitoxin activities would be encoded within the same catalytic site under the control of regulatory metals. They would therefore differ from type III TA modules, in which the toxin is repressed by direct interaction with the antitoxin, or type IV TA modules, in which the effects of the toxin are reverted by the antitoxin without direct interaction (reviewed in ref. [28]). It is interesting to note that the antitoxin component of TA modules such as Bartonella VbhT/ VbhA features a structurally equivalent glutamate that points into the catalytic site of the toxin[8], raising the question of whether there may exist conditions where the antitoxin glutamate implements deAMPylation within the TA complex, possibly in a metal-dependent mechanism. The physiological conditions that yield variations in metal homeostasis to activate the switch between opposing activities in these FIC proteins and how individual bacteria exploit the bifunctionality of their FIC proteins within their ecological niche are important issues to address in future studies. Potential conditions where bifunctional FIC proteins may contribute to virulence control could be the adaptation of pathogen growth to changes in Mg$^{2+}$ homeostasis (reviewed in ref. [26]) or the response of bacteria to Ca$^{2+}$ influx upon loss of bacterial wall integrity, a situation that may be encountered by enterobacteria in the gut where Ca$^{2+}$ concentrations are high.

Finally, our discovery that deAMPylation of the BiP chaperone by human FICD is modulated by changes in Ca$^{2+}$ concentration raises important questions with respect to the regulation and role of FICD in ER functions. The ER is the major organelle involved in Ca$^{2+}$ homeostasis (reviewed in refs. [29–31]), and the free Ca$^{2+}$ concentration in the ER is as high as 2–3 mM under resting conditions[32]. Disruption of Ca$^{2+}$ homeostasis, such as depletion of the ER Ca$^{2+}$ store induced by the drug thapsigargin, swiftly alters protein folding processes and activates the UPR[33]. Large Ca$^{2+}$ fluctuations are therefore considered as major determinants of ER stress responses (reviewed in refs. [30,31]). On the other hand,

the free $Mg^{2+}$ concentration in cells is estimated to be in the 0.5–1 mM range, and there is currently no evidence that it undergoes large fluctuations (reviewed in ref. [34]). Thus, while keeping in mind that free $Ca^{2+}$ and $Mg^{2+}$ concentrations and local gradients in the ER have remained difficult to determine with accuracy (reviewed in ref. [34]), an essentially constant $Mg^{2+}$ concentration of about 1 mM and $Ca^{2+}$ fluctuations in the 10 μM–3 mM range, similar to those used in our vitro assays, are plausibly encountered in the ER as it experiences transition from resting to stress conditions. Recently, FICD has been demonstrated to stimulate the activity of the BiP chaperone in response to an increase in the unfolded protein load[13–15,19], and this activation was correlated to deAMPylation of BiP by FICD[18]. It is thus tempting to speculate that inhibition of FICD deAMPylation activity by high $Ca^{2+}$, which we observe in vitro, reflects its inhibition under ER homeostasis, where $Ca^{2+}$ concentration is high and BiP activity is low. Conversely, depletion of $Ca^{2+}$ induces ER stress and triggers the UPR. Depletion of $Ca^{2+}$ may suppress inhibition of FICD deAMPylation activity, leading to efficient deAMPylation of BiP and upregulation of its activity, which is a key feature of the UPR. We speculate that FICD has thus features of an enzymatic sensor of $Ca^{2+}$ in the ER that could allow it to function as an integrator between $Ca^{2+}$ homeostasis in the ER and the BiP-mediated UPR.

In conclusion, we have identified $Ca^{2+}$ as a diffusable signal that modulates the intrinsic enzymatic activity of glutamate-bearing FIC proteins directly and tips the balance between AMPylation and deAMPylation reactions without a conformational change in the enzyme. Future studies are now needed to investigate how the metal switch of glutamate-bearing FIC protein activities is exploited in bacterial stress, and, in the case of animal FICD, its role in the UPR and its crosstalks with $Ca^{2+}$-controlled processes in the ER. While designing mutants that affect the $Ca^{2+}$ response specifically might be difficult due to the intertwined AMPylation and deAMPylation catalytic sites, chemical biology approaches combining the use of ATP analogs and caged metals might provide alternatives for future investigations of the $Ca^{2+}$ switch in a cell context. The detrimental effects of prolonged UPR in a myriad of diseases (reviewed in refs. [35,36]) makes new druggable targets highly desirable (reviewed in ref. [37]). Given the established pivotal role of the BiP chaperone (reviewed in ref. [16]) and of calcium (reviewed in ref. [31]) in this process, the $Ca^{2+}$-regulated activity of FICD uncovered in this study may constitute such a druggable target.

## Methods

**Protein expression and purification.** The codon-optimized gene encoding full-length *Enterococcus faecalis* EfFIC with an N-terminal 6-histidine tag was synthesized by GeneArt Gene Synthesis (ThermoFisher Scientific) and cloned into a pET22b(+) vector. The codon-optimized gene encoding human FICD/HYPE (residues 45–459, DNA sequence given in Supplementary Table 2) carrying an N-terminal 6-His tag followed by SUMO tag was synthesized by GeneArt Gene Synthesis and cloned into a pET151/D-TOPO vector (ThermoFisher Scientific). All mutations were performed with the QuickChange II Mutagenesis Kit (Agilent). The primers for the EfFIC[E115A], EfFIC[E190G], and FICD[E234G] mutants are listed in Supplementary Table 3. The EfFIC[H111A] mutant was prepared by an institutional cloning facility, which closed in the meantime and we therefore have no means to recover primer information. However, the crystal structure of the EfFIC[H111A] mutant has been determined in this study, confirming the mutation.

*Mus musculus* BiP in pUJ4 plasmid is a kind gift from Ronald Melki (CNRS, Gif-sur-Yvette). All constructs were verified by sequencing (GATC). All EfFIC constructs were expressed in *E. coli* BL21 (DE3) pLysS in LB medium. Overexpression was induced overnight with 0.5 mM IPTG at 20 °C. Bacterial cultures were centrifuged for 40 min at 4000 × g. Bacterial pellets were resuspended in lysis buffer (50 mM Tris-HCl pH 8.0, 150 mM NaCl, 5% glycerol, 0.25 mg/mL lysozyme) containing a protease inhibitor cocktail, disrupted at 125 psi using a high pressure cell disrupter and centrifuged 30 min at 22,000 × g. The cleared lysate supernatant was loaded onto an Ni-NTA affinity chromatography column (HisTrap FF, GE Healthcare) and eluted with 250 mM imidazole. Purification was polished by gel filtration on a Superdex 200 16/600 column (GE Healthcare)

equilibrated with storage buffer (50 mM Tris-HCl pH 8.0, 150 mM NaCl). Wild-type and mutant FICD/HYPE were expressed and purified as EfFIC, except that the lysis buffer was supplemented with 1 mM dithiothreitol (DTT) and 0.02 % Triton X-100 and the other buffers were supplemented with 1 mM DTT. To remove the SUMO tag, FICD/HYPE was incubated with SUMO protease (ThermoFischer) at 1/100 weight/weight ratio during 1 h at room temperature. The cleaved fraction was separated by affinity chromatography (HisTrap FF, GE Healthcare) and further purified by gel filtration on a Superdex 200 10/300 column (GE Healthcare) equilibrated with storage buffer (50 mM Tris pH 8.0, 150 mM NaCl, 1 mM DTT, 5% glycerol). Mouse BiP was expressed and purified essentially as EfFIC.

All proteins used in this study are highly pure, as shown by sodium dodecyl sulfate-polyacrylamide gel electrophoresis (SDS-PAGE; Supplementary Fig. 4).

**Crystallization and structure determination.** A summary of the crystal structures determined in this study is in Table 1. Crystallization screens (Jena Bioscience and Quiagen) were carried out using a TTP Labtech's Mosquito LCP crystallization robot. Conditions leading to crystals were subsequently optimized. Diffraction data sets were recorded at synchrotron SOLEIL and ESRF. Data sets were processed using XDS[38], xdsme (https://github.com/legrandp/xdsme), or autoProc[39]. Structures were solved by molecular replacement and refined with the Phenix suite[40] or Buster[41]. Models were build using Coot[42]. Softwares used in this project were curated by SBGrid[43]. Crystallization conditions are in Supplementary Methods. Data collection statistics and refinement statistics are given in Supplementary Table 1. All structures have been deposited with the Protein Data Bank.

**AMPylation and deAMPylation assays.** AMPylation and deAMPylation autoradiography assays were carried out using the following protocols. For AMPylation reactions, purified proteins (8 μM) were mixed with [α-$^{32}$P] ATP (10 μCi) (Perkin Elmer) in a buffer containing 50 mM Tris-HCl pH 7.4, 150 mM NaCl, and 0.1 mM $MgCl_2$. Reactions were incubated for 1 h at 30 °C, then stopped by addition of reducing SDS sample buffer and boiling for 5 min. For deAMPylation, AMPylation was performed as above, then 1 mM EDTA was added with or without 10 mM $CaCl_2$. Proteins were resolved by SDS-PAGE and AMPylation was revealed by autoradiography.

EfFIC AMPylation and deAMPylation fluorescence assays were carried out using the following protocols. AMPylation was carried out using a fluorescent ATP analog modified by $N^6$-(6-Amino)hexyl on the adenine base (ATP-FAM, Jena Bioscience). AMPylated proteins were prepared by incubation with an equimolar amount of ATP-FAM for 1 h at 30 °C in 50 mM Tris pH 8.0, 150 mM NaCl, and 0.1 mM $MgCl_2$. Before deAMPylation reactions, the buffer was exchanged to 50 mM Tris-HCl pH 8.0 and 150 mM NaCl by 5 cycles of dilution/concentration on a Vivaspin-500 with a cut-off of 10 kDa (Sartorius), resulting in a final dilution of ATP-FAM, $MgCl_2$, and PPi by about $10^5$ times. DeAMPylation reactions were carried out using 3 μM of AMPylated protein and 6 μM of freshly purified EfFIC proteins except when indicated otherwise in a buffer containing 50 mM Tris-HCl pH 8.0 and 150 mM NaCl for 1 h at 30 °C. Reactions were stopped by addition of reducing SDS sample buffer and boiling for 5 min. Proteins were resolved by SDS-PAGE and modification by AMP-FAM was revealed by fluorescence using green channel (excitation: 488 nm, emission: 526 nm) on a Chemidoc XR+Imaging System using ImageLab (BioRad).

$K_D$ of $Mg^{2+}$ and $Ca^{2+}$ for the deAMPylation reaction were determined by measuring deAMPylation efficiencies over a range of metal concentration (5–1000 μM) using fluorescence. DeAMPylation of $^{AMP-FAM}$EfFIC[E190G] by EfFIC[WT] (4 μM) was carried out for 5 min at room temperature, then reactions were stopped as described above. DeAMPylation efficiencies are expressed as the percentage of deAMPylation measured for 1 h at 30 °C in the presence of 3 mM of the corresponding metal; 0% is set to the deAMPylation efficiency measured in the absence of metal. $K_D$s were obtained from the titration curves using PRISM 7 (GraphPad Software).

FICD AMPylation and deAMPylation fluorescence assays were carried out using the following protocols. AMPylation was carried out using fluorescent ATP-FAM. AMPylated BiP ($^{AMP-FAM}$BiP) was prepared by incubation with FICD[E234G] (2 μM) and an equimolar amount of ATP-FAM for 1 h at 30 °C in 50 mM Tris pH 8.0, 150 mM NaCl, and 0.1 mM $MgCl_2$. Before deAMPylation reactions, the buffer was exchanged with 50 mM Tris-HCl pH 8.0 and 150 mM NaCl by 5 cycles of dilution/concentration on a Vivaspin-500 with a cut-off of 50 kDa (Sartorius), resulting in a final dilution of ATP-FAM, $MgCl_2$, and PPi produced by the reaction of about $10^5$ times. DeAMPylation reactions were carried using $^{AMP-FAM}$BiP (2 μg) and freshly purified FICD[WT] (4 μg) in a buffer containing 50 mM Tris pH 8.0 and 150 mM NaCl for 1 h at 30 °C. Reactions were stopped and AMPylation levels were revealed and quantified as described for EfFIC above.

All experiments were done in duplicates or triplicates, as indicated in the figure legends. Statistics were performed using XLSTAT 2018.3.50819 with a two-sample $t$ test and $z$-test.

**Reporting summary.** Further information on experimental design is available in the Nature Research Reporting Summary linked to this article.

## Data availability

Coordinates and structure factors have been deposited in the Protein Data Bank under accession codes 6ER6, 5NV5, 6EP0, 6EP2, 6ERB, 5NWF, 6EP5. The uncropped gels underlying Figs. 1f, 2d, e, 3a, c, d, 4a–c and the reported averages in the graphs of Figs. 3a, d and 4c are provided as a Source Data file. A reporting summary for this Article is available as a Supplementary Information file. All other data supporting the findings of this study are available from the corresponding author on reasonable request.

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

## Acknowledgements

This work was supported by grants from the Fondation pour la Recherche Médicale (grant number DEQ20150331694) and from the Agence Nationale pour la Recherche (ANR-14-CE09-0028-01) to J.C. and by the Agence Nationale pour la Recherche (ANR-10-LABX-62-IBEID) and from the Fondation pour la Recherche Médicale to C.B. S.V. was supported by a PhD grant from the DIM MALINF and G.O. by a stipend from the Pasteur-Paris University International PhD program. We are grateful to the scientific teams at the PX1 and PX2 beamlines at the SOLEIL synchrotron (Gif-sur-Yvette, France) and from the ID29, ID30-A3, and ID30B beamlines at the European Synchrotron Research Facility (ESRF, Grenoble, France) for their expertise and advice. We thank Pascale Serror (INRA, Jouy-en-Josas, France) and Philippe Glaser (Institut Pasteur) for discussions and the members of the Cherfils laboratory for help and shared expertise.

## Author contributions

S.V. designed, performed, and analyzed all crystallographic and biochemical experiments; G.O., M.R., and C.B. performed and analyzed autoradiography experiments; G.P. supervised the research and analyzed data; J.C. conceived and supervised the research, analyzed data, and wrote the manuscript with input from S.V and G.P.

## Additional information

**Competing interests:** The authors declare no competing interests.

