## [Peer Review File · Nature Communications]

Reviewers' comments:

Reviewer #1 (Remarks to the Author):

The manuscripts by Veyron et al. provides new structural insights of particularly high and novel mechanistic value with respect to the control of AMPylation/de AMPylation by a single enzyme. The authors document a Mg²⁺ /Ca²⁺ switch in 2 FIC proteins, based on the presence of the conserved glutamate that represses AMPylation. Structures and in vitro data (PTM assays) are very clearly presented and described.

Ultimately, the importance of the proposed regulatory mechanism, more specifically the regulation of FICD by Ca²⁺ during the UPR, will depend on further studies in relevant cellular contexts and conditions. Although some points have been described in the discussion, further aspects of physiological significance should be considered and included. The apparent problem of much higher concentrations of cellular Mg²⁺ (15-20mM) compared to Ca²⁺ should be addressed. Similarly, the authors should include comments related to generation of FIC variants, by structure-based mutations, that could test the Mg²⁺ /Ca²⁺ switch mechanism in a cellular context.

Reviewer #2 (Remarks to the Author):

Veyron et al report an enzymatic analysis of the FIC proteins EfFIC from *Enterococcus faecalis* and human FICD/HYPE and investigate the role of divalent cations (Mg⁺⁺, Ca⁺⁺) on their AMPylation and deAMPylation activity. For this they follow EfFIC auto-modification and for FICD mediated BiP target modification. Their studies are complemented by several EfFIC crystal structures complexed with various ligands.

Reading the abstract, the main claims of this manuscript are

(1) that “ ... EfFIC catalyzes both AMPylation and deAMPylation within the same active site, suggesting that the conserved glutamate is the signature of AMPylation/deAMPylation bifunctionality.”

(2) that they " ...identify a multi-position metal switch implemented by the glutamate, whereby EffIC uses Mg²⁺ and Ca²⁺ to control AMPylation and deAMPylation differentially without conformational change".

(3) and that " Remarkably, ... variations in Ca²⁺ levels also tune deAMPylation of BiP by human FICD"

Ad (1): EffIC is a close homolog (>50 % sequence identity) of the well characterised NmFIC enzyme from *N. meningitidis* for which GyrB target AMPylation as well as autoAMPylation has been demonstrated before (Stanger et al., 2016). So, the AMPylation activity of EffIC comes to no surprise. FIC mediated deAMPylation has been shown hitherto only for human FICD (Preissler et al., 2016). So, this is the first report of a bacterial FIC protein to catalyse deAMPylation. Given the conservation of the active site between human FICD and bacterial EffIC, it is most likely that the mechanism is the same. Indeed the mechanistic proposal given by Preissler et al. (Fig. 7) and by the authors (Fig. 5) is virtually identical with the important addition of a divalent cation based on the EffIC/AMP/CA⁺⁺ structure.

That presence/absence of the "conserved glutamate" (meaning inhibitory glutamate of the inhibitory helix) correlates with AMPylation and deAMPylation activity, respectively, has been shown very nicely in the Preissler paper.

Thus, the presented data show (not more and not less) that the AMPylation and deAMPylation mechanism and the regulatory role of the glutamate appears to be conserved from humans to bacteria.

Ad (2): The authors have extended the deAMPylation reaction scheme proposed by Preissler et al. by considering the role of a divalent cation, which would activate the hydrolytic water or may be important for transition state stabilization. This is based on (a) the observation of a calcium ion in the AMP complex that appears to be in a favorable location to coordinate with the hydrolytic water molecule in the required in-line position with the scissile bond) and (b) with the deleterious effect of point mutants that have either the coordinating E115 of the EffIC loop or the indirectly coordinating inhibitory glutamate truncated.

The basis for the additional claim that magnesium and calcium control differentially the two antagonistic activities of the bifunctional enzyme is, in my mind, rather weak and the notion that regulation may thus be achieved without conformational change (movement of the inhibitory glutamate) very questionable.

Ad (3): That replacement of a magnesium cofactor by calcium is deleterious for enzyme activity is surely not unusual and probably due to altered coordination distances affecting the position of reacting groups (here the position of a hydrolytic water). The authors show calcium mediated inhibition of FICD catalysed deAMPylation, but apparently have not studied the effect of the metal on AMPylation (which most likely is also inhibitory as observed for EcFIC).

Do the authors challenge the model that FICD AMPylation/deAMPylation is controlled by movement of the inhibitory helix carrying the inhibitory glutamate (and induced in an unknown way by an input signal) as suggested by Preissler et al. (Fig. 7) and want to suggest that FICD is solely regulated by differential calcium mediated inhibition of AMPylation and deAMPylation activity? I don't see how in such a scenario auto-inhibition of wt FICD AMPylation activity could be overcome. If, on the other hand, the authors merely want to suggest that deAMPylation proceeds with distinct speed in presence of magnesium vs calcium, the physiological relevance of such a response (e.g. to a cellular calcium spike) may be rather minor. Probably, the activity of most magnesium activated enzymes will be altered by competing calcium.

The investigation of the calcium effect on EffIC and FICD would have to be improved considerably to be able to judge the relevance of the reported observations. This would involve

Determination of the kinetics of AMPylation and deAMPylation under the various conditions, i.e. acquisition of progress curves and calculation of k_{cat} values. Experiments have to be done at known ATP concentrations (spiked with hot ATP), not just with the minute (unknown?) amount of hot ATP. End point measurements are qualitative at best.

DeAMPylation measurements with pre-autoAMPylated and purified EffIC, to avoid complex kinetics for the occurrence of the AMPylated species (reaction intermediate), when using a mixture of AMPylating and deAMPylating enzymes (EfFIC(EtoG) and wt EffIC).

Divalent cation (Mg^{+} , Ca^{++}) titration to obtain the corresponding activation constants (affinity of the metal), and k_{cats} for the AMPylation and deAMPylation reactions. Only with such quantitative knowledge the AMPylation state of the protein under given conditions can be predicted (simulated).

Several crystal structures are presented. The most relevant structure is clearly that of EffIC/AMP/ Ca^{++} , but its representation in Fig. 2B is very terse. To understand the structure, a figure

with all atoms of the binding site has to be shown. What is the relation of the ligands and the proposed hydrolytic water to the N-term of the underlying helix? What are the individual coordination distances with the calcium ion ("Ca²⁺ has 6 coordinations with distances in the expected 2.1-2.9 Å range" is not sufficient)? What are the B-factors of the ligating groups? (The PDB report for 6ep0 gives a rather large B-value of 64 for the metal, and an outstanding large LLDF {"quality of electron density of the group with respect to its neighbouring residues"}, in contrast to the calcium in the 6ep2 structure.) Is the calcium fully occupied? Most importantly, is the proposed hydrolytic water exactly in-line (give the angle)? How would the geometry be different with a magnesium ion (do the respective model)?

The authors claim that ATP_{gamma}S binds in a competent orientation to wild-type EffIC (Fig. 1E). Are the authors sure that the gamma-phosphate is still present? Do the authors suggest that ATP would bind also competently to EffIC? Why then such a strong boost of activity (to be quantified (!)) upon the E to G mutation? In Fig. 1D, the wtEffIC/ATP_{gamma}S structure is compared with wtNmFic/AMPPNP (NOT ATP_{gamma}S as labeled), 3s6a). For a meaningful comparison the same non-hydrolyzable ligand should be used and the presence of the gamma-phosphate should be verified.

It is somewhat amazing that the authors give so little information about their EffIC structures and their oligomeric forms. I checked the 5nv5 structure, which is already available, and it turns out that it is the same tetramer as NmFic (Stanger et al., 2016), stabilised by homologous salt-bridges. Also, they discuss the complex regulatory mechanism of the close NmFic homolog reported in this paper only very tersely. This is again amazing, since they show (only qualitative) data about the protein concentration dependence of EffIC mediated de-AMPylation (Fig. 3a).

Similarly, other related FIC knowledge is cited in a rather sketchy way. What is known about autophosphorylation, which site gets modified, how does this contribute to activation, what is the role of oligomerization? For an overview, see e.g. Casey and Orth, 2017.

Summarizing, this study shows that a bacterial FIC protein can catalyse deAMPylation. Furthermore, it points to the involvement of a divalent cation in this activity, but otherwise does not give any substantial or sound insight into the corresponding mechanism. A systematic, quantitative investigation is needed to advance significantly our knowledge about this antagonistic activity of FIC enzymes.

Responses to reviewers.

Reviewer 1

Comments. The manuscript by Veyron et al. provides new structural insights of particularly high and novel mechanistic value with respect to the control of AMPylation/de AMPylation by a single enzyme. The authors document a Mg²⁺ /Ca²⁺ switch in 2 FIC proteins, based on the presence of the conserved glutamate that represses AMPylation. Structures and in vitro data (PTM assays) are very clearly presented and described.

Ultimately, the importance of the proposed regulatory mechanism, more specifically the regulation of FICD by Ca²⁺ during the UPR, will depend on further studies in relevant cellular contexts and conditions. Although some points have been described in the discussion, further aspects of physiological significance should be considered and included. The apparent problem of much higher concentrations of cellular Mg²⁺ (15-20mM) compared to Ca²⁺ should be addressed. Similarly, the authors should include comments related to generation of FIC variants, by structure-based mutations, that could test the Mg²⁺ /Ca²⁺ switch mechanism in a cellular context.

Response. We thank this reviewer for their positive comments. We agree with this reviewer that a key aspect in the proposed regulatory mechanism is the concentrations of free metal ions. We have therefore reworded our discussion to make this point clearer and to emphasize that the ability of cells to use metal ions for regulation depends on their free concentrations (lines 335-342). Specifically, in the case of Mg²⁺, while the total concentration is indeed in the 15-20 mM range, the free concentration is in the 0.5-1 mM range in various cell types (reviewed in Romani Arch Biochem Biophys 2011). Furthermore, our thorough analysis of the literature gives no indication that the free concentration of Mg²⁺ fluctuates in the ER, keeping in mind that this concentration has remained difficult to establish notably because the high concentration of Ca²⁺ interferes with its precise measurement (reviewed in Romani 2011). The concentration of free Ca²⁺ in the ER is as high as 2-3 mM under resting condition (Montero et al., EMBO J. 2015). Thus, the concentrations of Mg²⁺ and Ca²⁺ used in our assays fall within the ranges of their free concentrations in cells.

We agree with this reviewer that it will be important for future studies to design experimental strategies that can test the metal switch mechanism in a cellular context. It is likely that mutations affecting Ca²⁺/Mg²⁺ binding or the AMPylation or deAMPylation reactions selectively, will be extremely difficult to design because the AMPylation and deAMPylation catalytic sites and the metal-binding sites are intertwined and the exact same amino acids are critical for both reactions and binding sites. As an alternative, we suggest chemical biology approaches using caged metals with photoactive chelators in combination with nucleotide analogs, which can be released into bacteria and cellular compartments in a controlled manner. We have added a comment to suggest directions for future studies (lines 359-363).

Reviewer 2

Veyron et al report an enzymatic analysis of the FIC proteins EffFIC from Enterococcus faecalis and human FICD/HYPE and investigate the role of divalent cations (Mg⁺⁺, Ca⁺⁺) on their AMPylation and deAMPylation activity. For this they follow EffFIC auto-modification

and for FICD mediated BiP target modification. Their studies are complemented by several EffFIC crystal structures complexed with various ligands.

Reading the abstract, the main claims of this manuscript are

- (1) that “ ... EffFIC catalyzes both AMPylation and deAMPylation within the same active site, suggesting that the conserved glutamate is the signature of AMPylation/deAMPylation bifunctionality.”
- (2) that they “ ...identify a multi-position metal switch implemented by the glutamate, whereby EffFIC uses Mg²⁺ and Ca²⁺ to control AMPylation and deAMPylation differentially without conformational change”.
- (3) and that “ Remarkably, ... variations in Ca²⁺ levels also tune deAMPylation of BiP by human FICD“□

Ad (1): EffFIC is a close homolog (>50 % sequence identity) of the well characterised NmFIC enzyme from N. meningitidis for which GyrB target AMPylation as well as autoAMPylation has been demonstrated before (Stanger et al., 2016). So, the AMPylation activity of EffFIC comes to no surprise. FIC mediated deAMPylation has been shown hitherto only for human FICD (Preissler et al., 2016). So, this is the first report of a bacterial FIC protein to catalyse deAMPylation. Given the conservation of the active site between human FICD and bacterial EffFIC, it is most likely that the mechanism is the same. Indeed the mechanistic proposal given by Preissler et al. (Fig. 7) and by the authors (Fig. 5) is virtually identical with the important addition of a divalent cation based on the EffFIC/AMP/CA⁺⁺ structure.

That presence/absence of the “conserved glutamate” (meaning inhibitory glutamate of the inhibitory helix) correlates with AMPylation and deAMPylation activity, respectively, has been shown very nicely in the Preissler paper. Thus, the presented data show (not more and not less) that the AMPylation and deAMPylation mechanism and the regulatory role of the glutamate appears to be conserved from humans to bacteria.

Response: The glutamate was coined « inhibitory » before the recent discovery that it plays a critical role in the deAMPylation reaction in human FICD (Preisler et al., NSMB 2017) and in bacterial FIC proteins (this study). For this reason, we find it more appropriate to refer to it as the « conserved » glutamate in our manuscript. We explain our choice at lines 169-171 and we have made sure that there is no ambiguity that we refer to the inhibitory glutamate of the inhibitory helix.

Ad (2): The authors have extended the deAMPylation reaction scheme proposed by Preissler et al. by considering the role of a divalent cation, which would activate the hydrolytic water or may be important for transition state stabilization. This is based on (a) the observation of a calcium ion in the AMP complex that appears to be in a favorable location to coordinate with the hydrolytic water molecule in the required in-line position with the scissile bond) and (b) with the deleterious effect of point mutants that have either the coordinating E115 of the EffFIC loop or the indirectly coordinating inhibitory glutamate truncated.

The basis for the additional claim that magnesium and calcium control differentially the two antagonistic activities of the bifunctional enzyme is, in my mind, rather weak and the notion that regulation may thus be achieved without conformational change (movement of the inhibitory glutamate) very questionable.

Response: Our proposed metal-based regulatory mechanism is cross-validated by several distinct biochemical assays, using not only mutants but also wild-type enzymes, and by several crystal structures. We are including a new experiment that further validates this mechanism. In this experiment, we analyzed the kinetics of the net AMPylation by EffFIC in response to a Ca^{2+} increase. This kinetics assay shows that EffFIC is able to switch quickly from the AMPylation regime to the deAMPylation regime in response to the addition of Ca^{2+} to the enzyme pre-loaded with Mg^{2+} . The experiment is described lines 206-209 and is shown in a new panel in Figure 3 (3E).

Ad (3): That replacement of a magnesium cofactor by calcium is deleterious for enzyme activity is surely not unusual and probably due to altered coordination distances affecting the position of reacting groups (here the position of a hydrolytic water).

Response : The fact that Ca^{2+} may affect unrelated enzymes in an artifactual manner cannot be used as an argument to dismiss its effect on FIC proteins. Importantly, while further studies in cellular contexts are certainly needed, we emphasize that calcium is a primary modulator of ER functions and that it undergoes large fluctuations in the ER. Hence FICD does experience Ca^{2+} fluctuations, which makes it physiologically plausible that it perceives, and responds to, Ca^{2+} fluxes. Bacteria such as *E. faecalis*, which are found in the gut where levels of free Ca^{2+} are high, may also plausibly experience changes in Ca^{2+} homeostasis, although this remains to be investigated.

The authors show calcium mediated inhibition of FICD catalysed deAMPylation, but apparently have not studied the effect of the metal on AMPylation (which most likely is also inhibitory as observed for EcFIC).

Response. The effect of metals on AMPylation of BiP by FICD is shown in Figure 4A (unchanged from the previous manuscript version). We did not detect BiP AMPylation by wild-type FICD using fluorescence, regardless of whether the metal is Mg^{2+} or Ca^{2+} . This is consistent with previous observations using autoradiographies that the AMPylation activity of wild-type FICD is repressed by the inhibitory glutamate (for example, Ham et al., JBC 2014, Engel et al., Nature 2012).

Do the authors challenge the model that FICD AMPylation/deAMPylation is controlled by movement of the inhibitory helix carrying the inhibitory glutamate (and induced in an unknown way by an input signal) as suggested by Preissler et al. (Fig. 7) and want to suggest that FICD is solely regulated by differential calcium mediated inhibition of AMPylation and deAMPylation activity? I don't see how in such a scenario auto-inhibition of wt FICD AMPylation activity could be overcome. If, on the other hand, the authors merely want to suggest that deAMPylation proceeds with distinct speed in presence of magnesium vs calcium, the physiological relevance of such a response (e.g. to a cellular calcium spike) may be rather minor.

Response : Multiple layers of regulations are classically found in many regulatory proteins. The fact that EffFIC and FICD activities are modulated by metals is not incompatible with these enzymes having other layers of regulations, such as autoAMPylation or changes in oligomerization, and we mention this again at lines 303-306 in the revised version. However, as this reviewer points out, physiological signals lifting autoinhibition of AMPylation by the inhibitory helix have remained unknown.

Probably, the activity of most magnesium activated enzymes will be altered by competing calcium.

Response : please see our response to this statement above.

The investigation of the calcium effect on EffIC and FICD would have to be improved considerably to be able to judge the relevance of the reported observations. [...] Only with such quantitative knowledge the AMPylation state of the protein under given conditions can be predicted (simulated).

This would involve

Determination of the kinetics of AMPylation and deAMPylation under the various conditions, i.e. acquisition of progress curves and calculation of k_{cat} values. Experiments have to be done at known ATP concentrations (spiked with hot ATP), not just with the minute (unknown?) amount of hot ATP. End point measurements are qualitative at best.

Response : The experiments we are carrying out throughout this study are not end-point measurements, but two-point measurements, which allow to quantify variations in either the substrate (for deAMPylation) or the product (for AMPylation) between $t=0$ and a given time. This experimental approach is classically used, notably to analyze the kinetics of post-translational modifications. Autoradiography experiments are carried out at fixed amounts of radioactive ATP (expressed in radioactivity units), which makes our experiments fully reproducible and comparable. Importantly, our observations obtained with radioactive ATP are cross-validated with fluorescent ATP-FAM.

DeAMPylation measurements with pre-autoAMPyated and purified EffIC, to avoid complex kinetics for the occurrence of the AMPylated species (reaction intermediate), when using a mixture of AMPylating and deAMPyating enzymes (EffIC(EtoG) and wt EffIC).

Response : We do not understand this comment. Actually, most our deAMPylation experiments are carried out with purified, pre-AMPyated proteins. Please see for example lines 156-159 and Figure 2E (EffIC), or lines 237-239 and Figure 4B (FICD and BiP).

Divalent cation (Mg^{++} , Ca^{++}) titration to obtain the corresponding activation constants (affinity of the metal), and k_{cats} for the AMPylation and deAMPylation reactions.

Response : We agree with this reviewer that the knowledge of the dissociation constants of the metals provides additional insight into the physiological relevance of our findings. We have added new experiments to measure the dissociation constant of Mg^{2+} and Ca^{2+} for the deAMPyating-competent form of EffIC. We find that the dissociation constants are in the same range for Ca^{2+} and Mg^{2+} , and that they are in the same range as dissociation constants reported for bacterial Mg^{2+} transporters. These observations further support the physiological relevance of the regulatory role of these metals. The experiment is described lines 197-200, in a new panel (3B) and in the methods (lines 440-447).

Several crystal structures are presented. The most relevant structure is clearly that of EffIC/AMP/ Ca^{++} , but its representation in Fig. 2B is very terse. To understand the structure, a figure with all atoms of the binding site has to be shown. What is the relation of the ligands and the proposed hydrolytic water to the N-term of the underlying helix? What are the individual coordination distances with the calcium ion ("Ca²⁺ has 6 coordinations with

distances in the expected 2.1-2.9 Å range” is not sufficient)? What are the B-factors of the ligating groups? (The PDB report for 6ep0 gives a rather large B-value of 64 for the metal, and an outstanding large LLDF {“quality of electron density of the group with respect to its neighbouring residues”}, in contrast to the calcium in the 6ep2 structure.) Is the calcium fully occupied? Most importantly, is the proposed hydrolytic water exactly in-line (give the angle)? How would the geometry be different with a magnesium ion (do the respective model)?

Responses: For clarity, we would prefer to keep our figures the way we prepared them originally, in which all relevant information discussed in the manuscript is shown in a manner that the figure is not overcrowded and is easy to understand. In the case of Figure 2B, we are showing all interactions of the calcium ion within the active site, corresponding to what we are discussing in the manuscript. Adding distance labels on each interaction or displaying amino acids that are not involved in this interaction would make this figure more difficult to read. The information requested by this reviewer is readily available from the PDB coordinates, which will be released upon publication at the latest.

This reviewer is correct that the 6ep0 structure features a calcium ion with a high B factor. However, this calcium is located in a crystal contact and is distinct from the calcium ion located in the active site which is discussed in the manuscript. To make that point clearer, we have included an omit map of the active site calcium ion from the EfFIC-Ca²⁺-ADP structure, which shows the good quality of the electron density map (Supplementary Figure 1C).

The position of the nucleophilic water molecule is modelled in Figure 2B (in grey, already shown in the previous version of the manuscript). We have reworded the description of the heptahedral geometry (lines 142-144) and the discussion on the position of the nucleophilic water (lines 282-285) to make these points clearer .

The authors claim that ATPgammaS binds in a competent orientation to wild-type EfFIC (Fig. 1E). Are the authors sure that the gamma-phosphate is still present? Do the authors suggest that ATP would bind also competently to EfFIC? Why then such a strong boost of activity (to be quantified (!)) upon the E to G mutation? In Fig. 1D, the wtEfFIC/ATPgammaS structure is compared with wtNmFiC/AMPPNP (NOT ATPgammaS as labeled), 3s6a). For a meaningful comparison the same non-hydrolyzable ligand should be used and the presence of the gamma-phosphate should be verified.

Response : We agree with this reviewer that since the gamma phosphate cannot be located in the electron density, the possibility that it is not present cannot formally be ruled out, similar to previous structures of NmMIC using AMPPNP (e.g. Engel et al., Nature 2010). However, we were unable to obtain crystals of EfFIC in the exact same conditions except that ATPγS was replaced by ADP, which would be in favor of ATPγS being the actual ligand. Note also that our analysis is focused on the cleavable bond located between the alpha and beta phosphates of the nucleotide, which is fully defined in the electron density and is chemically identical in ATP, AMPPNP and ATPγS.

It is somewhat amazing that the authors give so little information about their EfFIC structures and their oligomeric forms. I checked the 5nv5 structure, which is already available, and it turns out that it is the same tetramer as NmFic (Stanger et al., 2016), stabilised by homologous salt-bridges.

Response : We have included the requested analysis of the oligomeric states of EfFIC in all crystallographic structures (Figure S2, panels A-C and lines 179-186). Briefly, EfFIC forms

dimers in all crystal forms, but the arrangement of these dimers differs between crystals. Specifically, depending on the crystal forms, EffFIC does not form assemblies larger than the dimers, or forms tetramers related to the NmFIC tetramer, or forms hexamers comprised of trimers of dimers. Thus, the oligomeric state of EffFIC cannot be established based on the crystallographic assemblies and it is not always a tetramer. Please note that the crystallographic assemblies of all crystal structures were listed in Supplementary Table 1 (crystallographic summary) in our first submission.

Also, they discuss the complex regulatory mechanism of the close NmFic homolog reported in this paper only very tersely. This is again amazing, since they show (only qualitative) data about the protein concentration dependence of EffFIC mediated de-AMPylation (Fig. 3a).

Response : We do not understand this comment, as we are discussing the regulatory mechanism of NmFIC in the introduction (now lines 69-72, which we have slightly reworded to make it clearer), lines 176-179 in the results, and lines 303-306 in the discussion. Also, our analysis of the protein concentration dependency of EffFIC de-AMPylation (now moved to Supplementary Figure 2D) is not solely qualitative. Although we have not determined the quaternary structure that predominate at each concentration, the quantification clearly shows that deAMPylation increases with EffFIC concentration, hence that deAMPylation is not adversely affected by the toxin concentration.

Similarly, other related FIC knowledge is cited in a rather sketchy way. What is known about autophosphorylation, which site gets modified, how does this contribute to activation, what is the role of oligomerization? For an overview, see e.g. Casey and Orth, 2017.

Response : We have added a reference to the Casey and Orth 2017 review (lines 303-306).

Summarizing, this study shows that a bacterial FIC protein can catalyse deAMPylation. Furthermore, it points to the involvement of a divalent cation in this activity, but otherwise does not give any substantial or sound insight into the corresponding mechanism. A systematic, quantitative investigation is needed to advance significantly our knowledge about this antagonistic activity of FIC enzymes.

Response : We agree with this and the other reviewer that it will be important for future studies to design experimental strategies that can test the Mg^{2+}/Ca^{2+} switch mechanism in a cellular context. We propose directions for such studies in the revised manuscript. Notably, we included a comment to discuss physiological situations where bifunctional FIC proteins may contribute to virulence, such as adaptation to changes in Mg^{2+} homeostasis or loss of bacterial wall integrity leading to Ca^{2+} influx (lines 322-326). We also included a comment to discuss possible experimental strategies using chemical biology that could be used in future studies to address the metal switch in a cell context (lines 359-363).

Reviewers' comments:

Reviewer #1 (Remarks to the Author):

The revised manuscript addressed all my points of criticism

Reviewer #2 (Remarks to the Author):

Veyron et al. 2018 - revision

The authors have addressed few of my concerns, but do not accept my main criticism in that (i) a quantitative investigation is needed to characterise the individual functions of bifunctional FIC proteins and their regulation by divalent cations and (ii) that the biological relevance of the observed qualitative difference of the metals on the activities is purely speculative and as such merits a remark in the discussion, but not more.

In my mind, the value of the study lies in the demonstration of deAMPylation activity for a bacterial FIC protein and that this reaction requires a divalent metal. Furthermore, a crystal structure with AMP as a surrogate for an AMPylated target is presented that has proven useful for the discussion of the catalytic mechanism (position and activation of the hydrolytic water).

Summarizing, I cannot recommend this study for publication in a general, high-profile journal, but would recommend submission to a specialised journal.

—

I went through the rebuttal of the authors in detail and show my response below.

The authors show calcium mediated inhibition of FICD catalysed deAMPylation, but apparently have not studied the effect of the metal on AMPylation (which most likely is also inhibitory as observed for EcFIC).

Response. The effect of metals on AMPylation of BiP by FICD is shown in Figure 4A (unchanged from the previous manuscript version). We did not detect BiP AMPylation by wild-type FICD using fluorescence, regardless of whether the metal is Mg²⁺ or Ca²⁺. This is consistent with previous observations using autoradiographies that the AMPylation activity of wild-type FICD is repressed by the inhibitory glutamate (for example, Ham et al., JBC 2014, Engel et al., Nature 2012).

Response_response. I agree with the explanation why FICD is incompetent for AMPylation and, thus, cannot be tested for metal dependency. However, it is puzzling that autoAMPylation occurs (AMP-FAM_FIC(wt)), but not target modification. Is there an explanation? So, the metal dependency of autoAMPylation could be tested, or of the E->G mutant.

Do the authors challenge the model that FICD AMPylation/deAMPylation is controlled by movement of the inhibitory helix carrying the inhibitory glutamate (and induced in an unknown way by an input signal) as suggested by Preissler et al. (Fig. 7) and want to suggest that FICD is solely regulated by differential calcium mediated inhibition of AMPylation and deAMPylation activity? I don't see how in such a scenario auto-inhibition of wt FICD AMPylation activity could be overcome. If, on the other hand, the authors merely want to suggest that deAMPylation proceeds with distinct speed in presence of magnesium vs calcium, the physiological relevance of such a response (e.g. to a cellular calcium spike) may be rather minor.

Response : Multiple layers of regulations are classically found in many regulatory proteins. The fact that EffIC and FICD activities are modulated by metals is not incompatible with these enzymes having other layers of regulations, such as autoAMPylation or changes in oligomerization, and we mention this again at lines 303-306 in the revised version. However, as this reviewer points out, physiological signals lifting autoinhibition of AMPylation by the inhibitory helix have remained unknown.

Response_response. So, the authors agree that differential metal dependency of the two reactions would constitute a second layer of regulation. Would be appropriate to state that more clearly in the MS.

The investigation of the calcium effect on EffIC and FICD would have to be improved considerably to be able to judge the relevance of the reported observations. [...] Only with such quantitative knowledge the AMPylation state of the protein under given conditions can be predicted (simulated).

This would involve Determination of the kinetics of AMPylation and deAMPylation under the various conditions, i.e. acquisition of progress curves and calculation of k_{cat} values. Experiments have to be done at known ATP concentrations (spiked with hot ATP), not just with the minute (unknown?) amount of hot ATP. End point measurements are qualitative at best.

Response : The experiments we are carrying out throughout this study are not end-point measurements, but two-point measurements, which allow to quantify variations in either the substrate (for deAMPylation) or the product (for AMPylation) between $t=0$ and a given time. This experimental approach is classically used, notably to analyze the kinetics of post- translational modifications. Autoradiography experiments are carried out at fixed amounts of radioactive ATP (expressed in radioactivity units), which makes our experiments fully reproducible and comparable. Importantly, our observations obtained with radioactive ATP are cross-validated with fluorescent ATP-FAM.

Response_response. Yes, after a certain (arbitrary) incubation time the measurements were performed in a comparative fashion (Never heard about “2-point measurements”). But the pit-falls of such measurements should be obvious, especially for an enzyme with two antagonistic activities. The results will depend on the duration of incubation, in particular for the netAMPylation measurements (Figs.3D,E). Do the authors observe the initial phase, the steady state phase, or a late phase of the progress curve?

—

DeAMPylation measurements with pre-autoAMPyated and purified EffIC, to avoid complex kinetics for the occurrence of the AMPylated species (reaction intermediate), when using a mixture of AMPylating and deAMPyating enzymes (EfFIC(EtoG) and wt EffIC).

Response: We do not understand this comment. Actually, most our deAMPylation experiments are carried out with purified, pre-AMPyated proteins. Please see for example lines 156-159 and Figure 2E (EfFIC), or lines 237-239 and Figure 4B (FICD and BiP).

Response_response. I was wrong, sorry. This is fine.

Divalent cation (Mg^{++} , Ca^{++}) titration to obtain the corresponding activation constants (affinity of the metal), and k_{cat} s for the AMPylation and deAMPylation reactions.

Response : We agree with this reviewer that the knowledge of the dissociation constants of the metals provides additional insight into the physiological relevance of our findings. We have added new experiments to measure the dissociation constant of Mg^{2+} and Ca^{2+} for the deAMPylation-competent form of EfFIC. We find that the dissociation constants are in the same range for Ca^{2+} and Mg^{2+} , and that they are in the same range as dissociation constants reported for bacterial Mg^{2+} transporters. These observations further support the physiological relevance of the regulatory role of these metals. The experiment is described lines 197-200, in a new panel (3B) and in the methods (lines 440-447).

Response_response. Good to see the titrations and the derived affinities. But no estimates about (at least relative) k_{cat} values were derived, but data were normalised.

Several crystal structures are presented. The most relevant structure is clearly that of EfFIC/AMP/ Ca^{++} , but its representation in Fig. 2B is very terse. To understand the structure, a figure with all atoms of the binding site has to be shown. What is the relation of the ligands and the proposed hydrolytic water to the N-term of the underlying helix? What are the individual coordination distances with the calcium ion (“ Ca^{2+} has 6 coordinations with distances in the expected 2.1-2.9 Å range” is not sufficient)? What are the B-factors of the ligating groups? (The PDB report for 6ep0 gives a rather large B-value of 64 for the metal, and an outstanding large LLDF {“quality of electron density of the group with respect to its neighbouring residues”}, in contrast to the calcium in the 6ep2 structure.) Is the calcium fully occupied? Most importantly, is the proposed hydrolytic water exactly in-line (give the angle)? How would the geometry be different with a magnesium ion (do the respective model)?

Responses: For clarity, we would prefer to keep our figures the way we prepared them originally, in which all relevant information discussed in the manuscript is shown in a manner that the figure is not overcrowded and is easy to understand. In the case of Figure 2B, we are showing all interactions of the calcium ion within the active site, corresponding to what we are discussing in the manuscript. Adding distance labels on each interaction or displaying amino acids that are not involved in this

interaction would make this figure more difficult to read. The information requested by this reviewer is readily available from the PDB coordinates, which will be released upon publication at the latest.

This reviewer is correct that the 6ep0 structure features a calcium ion with a high B factor. However, this calcium is located in a crystal contact and is distinct from the calcium ion located in the active site which is discussed in the manuscript. To make that point clearer, we have included an omit map of the active site calcium ion from the Effic-Ca²⁺-ADP structure, which shows the good quality of the electron density map (Supplementary Figure 1C).

The position of the nucleophilic water molecule is modelled in Figure 2B (in grey, already shown in the previous version of the manuscript). We have reworded the description of the heptahedral geometry (lines 142-144) and the discussion on the position of the nucleophilic water (lines 282-285) to make these points clearer .

Response_response. Still would be interested to see whether the calcium with its coordination sphere does not make any other contacts, e.g. to the backbone. Thank you for Fig. S1C and the clarifying text. Heptahedral coordination is new to me, in the provided reference I can't find an explanation. Probably the authors mean 6-fold coordination with one ligand missing and one coordination site interacting with 2 atoms? *italic*

—

The authors claim that ATP γ S binds in a competent orientation to wild-type Effic (Fig. 1E). Are the authors sure that the gamma-phosphate is still present? ***Do the authors suggest that ATP would bind also competently to Effic? Why then such a strong boost of activity (to be quantified (!)) upon the E to G mutation?*** In Fig. 1D, the wtEffic/ATP γ S structure is compared with wtNmFic/AMPPNP (NOT ATP γ S as labeled), 3s6a). For a meaningful comparison the same non-hydrolyzable ligand should be used and the presence of the gamma-phosphate should be verified.

Response : We agree with this reviewer that since the gamma phosphate cannot be located in the electron density, the possibility that it is not present cannot formally be ruled out, similar to previous structures of NmMIC using AMPPNP (e.g. Engel et al., Nature 2010). **However, we were unable to obtain crystals of Effic in the exact same conditions except that ATP γ S was replaced by ADP, which would be in favor of ATP γ S being the actual ligand.** Note also that our analysis is focused on the cleavable bond located between the alpha and beta phosphates of the nucleotide, which is fully defined in the electron density and is chemically identical in ATP, AMPPNP and ATP γ S.

Response_response. Not fully answered (see in bold italics). The argument in bold is not strong.

It is somewhat amazing that the authors give so little information about their EffFIC structures and their oligomeric forms. I checked the 5nv5 structure, which is already available, and it turns out that it is the same tetramer as NmFic (Stanger et al., 2016), stabilised by homologous salt-bridges.

Response : We have included the requested analysis of the oligomeric states of EffFIC in all crystallographic structures (Figure S2, panels A-C and lines 179-186). Briefly, EffFIC forms dimers in all crystal forms, but the arrangement of these dimers differs between crystals. Specifically, depending on the crystal forms, EffFIC does not form assemblies larger than the dimers, or forms tetramers related to the NmFIC tetramer, or forms hexamers comprised of trimers of dimers. Thus, the oligomeric state of EffFIC cannot be established based on the crystallographic assemblies and it is not always a tetramer. Please note that the crystallographic assemblies of all crystal structures were listed in Supplementary Table 1 (crystallographic summary) in our first submission.

Response-response. Thanks for Fig. S2. So the questions arise what is the K_d of dimer formation and whether the dimer is catalytically competent. The increase in deAMPylation activity with enzyme concentration shown in Fig. S2D is no proof at all that the dimer is active. With increasing concentration the monomer fraction will monotonically increase always, even when dimers are formed. Not end concentrations are needed, but the dependence of $k_{cat} = v_{init}/E$ on enzyme concentration (is it constant or does it go down?) .

Also, they discuss the complex regulatory mechanism of the close NmFic homolog reported in this paper only very tersely. This is again amazing, since they show (only qualitative) data about the protein concentration dependence of EffFIC mediated de-AMPylation (Fig. 3a).

Response: We do not understand this comment, as we are discussing the regulatory mechanism of NmFIC in the introduction (now lines 69-72, which we have slightly reworded to make it clearer), lines 176-179 in the results, and lines 303-306 in the discussion. Also, our analysis of the protein concentration dependency of EffFIC de-AMPylation (now moved to Supplementary Figure 2D) is not solely qualitative. Although we have not determined the quaternary structure that predominate at

each concentration, the quantification clearly shows that deAMPylation increases with EffIC concentration, hence that deAMPylation is not adversely affected by the toxin concentration.

Response-response. Thanks for the improvement of the description of the NmFic mechanism. For concentration dependence, see my previous comment.

--

Response to reviewers.

Reviewer #2 (Remarks to the Author):

Veyron et al. 2018 - revision

The authors have addressed few of my concerns, but do not accept my main criticism in that (i) a quantitative investigation is needed to characterise the individual functions of bifunctional FIC proteins and their regulation by divalent cations and (ii) that the biological relevance of the observed qualitative difference of the metals on the activities is purely speculative and as such merits a remark in the discussion, but not more.

In my mind, the value of the study lies in the demonstration of deAMPylation activity for a bacterial FIC protein and that this reaction requires a divalent metal. Furthermore, a crystal structure with AMP as a surrogate for an AMPylated target is presented that has proven useful for the discussion of the catalytic mechanism (position and activation of the hydrolytic water).

Summarizing, I cannot recommend this study for publication in a general, high-profile journal, but would recommend submission to a specialised journal.

I went through the rebuttal of the authors in detail and show my response below.

The authors show calcium mediated inhibition of FICD catalysed deAMPylation, but apparently have not studied the effect of the metal on AMPylation (which most likely is also inhibitory as observed for EcFIC).

Previous response. The effect of metals on AMPylation of BiP by FICD is shown in Figure 4A (unchanged from the previous manuscript version). We did not detect BiP AMPylation by wild-type FICD using fluorescence, regardless of whether the metal is Mg²⁺ or Ca²⁺. This is consistent with previous observations using autoradiographies that the AMPylation activity of wild-type FICD is repressed by the inhibitory glutamate (for example, Ham et al., JBC 2014, Engel et al., Nature 2012).

Response_response. I agree with the explanation why FICD is incompetent for AMPylation and, thus, cannot be tested for metal dependency. However, it is puzzling that autoAMPylation occurs (AMP-FAM_FIC(wt)), but not target modification. Is there an explanation? So, the metal dependency of autoAMPylation could be tested, or of the E->G mutant.

New response. This analysis of the metal dependency of wild-type FICD autoAMPylation is already provided in the manuscript. Please see Figure 4A, which shows that FICD^{WT} autoAMPylation occurs with both Mg²⁺ (lane 1) and Ca²⁺ (lane 2). We have added a reference to Ref 14 (Ham et al. JBC 2014) (line 239), which shows in Figure 2B that wild-type drosophila FICD also has autoAMPylation activity and negligible AMPylation activity towards BiP. We are discussing this currently unexplained observation at lines 303-306.

Do the authors challenge the model that FICD AMPylation/deAMPylation is controlled by movement of the inhibitory helix carrying the inhibitory glutamate (and induced in an unknown way by an input signal) as suggested by Preissler et al. (Fig. 7) and want to suggest that FICD is solely regulated by differential calcium mediated inhibition of AMPylation and deAMPylation activity? I don't see how in such a scenario auto-inhibition of wt FICD AMPylation activity could be overcome. If, on the other hand, the authors merely want to suggest that deAMPylation proceeds with distinct speed in presence of magnesium vs calcium, the physiological relevance of such a response (e.g. to a cellular calcium spike) may be rather minor.

Previous response : Multiple layers of regulations are classically found in many regulatory proteins. The fact that EfFIC and FICD activities are modulated by metals is not incompatible with these enzymes having other layers of regulations, such as autoAMPylation or changes in oligomerization, and we mention this again at lines 303-306 in the revised version. However, as this reviewer points out, physiological signals lifting autoinhibition of AMPylation by the inhibitory helix have remained unknown.

Response_response. So, the authors agree that differential metal dependency of the two reactions would constitute a second layer of regulation. Would be appropriate to state that more clearly in the MS.

New response. We agree that the different mechanisms of regulation that have been described (metal switch, autoAMPylation, oligomerization changes) may combine as multiple regulatory layers, or they could operate under different physiological conditions. We have reworded the discussion to make that point clearer (lines 307-311).

The investigation of the calcium effect on EfFIC and FICD would have to be improved considerably to be able to judge the relevance of the reported observations. [...] Only with such quantitative knowledge the AMPylation state of the protein under given conditions can be predicted (simulated). This would involve □ Determination of the kinetics of AMPylation and deAMPylation under the various conditions, i.e. acquisition of progress curves and calculation of kcat values. Experiments have to be done at known ATP concentrations (spiked with hot ATP), not just with the minute (unknown?) amount of hot ATP. End point measurements are qualitative at best.

Previous response: The experiments we are carrying out throughout this study are not end-point measurements, but two-point measurements, which allow to quantify variations in either the substrate (for deAMPylation) or the product (for AMPylation) between $t=0$ and a given time. This experimental approach is classically used, notably to analyze the kinetics of post-translational modifications. Autoradiography experiments are carried out at fixed amounts of radioactive ATP (expressed in radioactivity units), which makes our experiments fully reproducible and comparable. Importantly, our observations obtained with radioactive ATP are cross-validated with fluorescent ATP-FAM.

Response_response. Yes, after a certain (arbitrary) incubation time the measurements were performed in a comparative fashion (Never heard about “2-point measurements”). But the pit-falls of such measurements should be obvious, especially for an enzyme with two antagonistic activities. The results will depend on the duration of incubation, in particular for the netAMPylation measurements (Figs.3D,E). Do the authors observe the initial phase, the steady state phase, or a late phase of the progress curve?

New response. Discontinuous assays in which the signal is measured at a specific time (also called stopped/single point assays when a single point is used), are textbook enzymology assays (see for example Price and Stevens, Copeland, Cornish-Bowden) and classical alternatives to continuous assays when such assays are difficult to setup technically. Their usage in quantitative comparison of post-translational modification efficiencies is nicely exemplified by a study by the Kirshner lab in Molecular Cell 2010 (PMID: 20347419). In this study, the authors quantified the net activity of an enzyme catalyzing a post-translational modification, human mitotic kinase CDK1, by varying the antagonistic activities of the Wee1 kinase and the CDC25 phosphatase and the concentration of regulatory cyclins A and B. Experiments monitoring the effect of individual or multiple regulatory components were all based on single point discontinuous assays (see Figures 1A and 2A in this article), in a manner similar to that used here for assessing the antagonistic effects of Ca^{2+} and Mg^{2+} concentrations. From these data, the authors could deduce the modalities of human CDK1 regulation.

DeAMPylation measurements with pre-autoAMPylated and purified EfFIC, to avoid complex kinetics

for the occurrence of the AMPylated species (reaction intermediate), when using a mixture of AMPylating and deAMPylating enzymes (E_fFIC(EtoG) and wt E_fFIC).

New response: We do not understand this comment. Actually, most our deAMPylation experiments are carried out with purified, pre-AMPylated proteins. Please see for example lines 156-159 and Figure 2E (E_fFIC), or lines 237-239 and Figure 4B (FICD and BiP).

Response_response. I was wrong, sorry. This is fine.

Divalent cation (Mg⁺⁺, Ca⁺⁺) titration to obtain the corresponding activation constants (affinity of the metal), and k_{cats} for the AMPylation and deAMPylation reactions.

Previous response : We agree with this reviewer that the knowledge of the dissociation constants of the metals provides additional insight into the physiological relevance of our findings. We have added new experiments to measure the dissociation constant of Mg²⁺ and Ca²⁺ for the deAMPylating-competent form of E_fFIC. We find that the dissociation constants are in the same range for Ca²⁺ and Mg²⁺, and that they are in the same range as dissociation constants reported for bacterial Mg²⁺ transporters. These observations further support the physiological relevance of the regulatory role of these metals. The experiment is described lines 197-200, in a new panel (3B) and in the methods (lines 440-447).

Response_response. Good to see the titrations and the derived affinities. But no estimates about (at least relative) k_{cat} values were derived, but data were normalised.

New response. Determination of k_{cat} values requires that measurements are carried out in conditions where the substrate is in excess over the enzyme. In the case of auto-modification reactions, these conditions cannot be setup because in such reactions the enzyme and the substrate are the same. However, we would like to point out again that efficiencies can be compared without k_{cat} determinations, as exemplified by the article by the Kirschner lab mentioned above.

Several crystal structures are presented. The most relevant structure is clearly that of E_fFIC/AMP/CA⁺⁺, but its representation in Fig. 2B is very terse. To understand the structure, a figure with all atoms of the binding site has to be shown. What is the relation of the ligands and the proposed hydrolytic water to the N-term of the underlying helix? What are the individual coordination distances with the calcium ion (“Ca²⁺ has 6 coordinations with distances in the expected 2.1-2.9 Å range” is not sufficient)? What are the B-factors of the ligating groups? (The PDB report for 6ep0 gives a rather large B-value of 64 for the metal, and an outstanding large LLDF {“quality of electron density of the group with respect to its neighbouring residues”}, in contrast to the calcium in the 6ep2 structure.) Is the calcium fully occupied? Most importantly, is the proposed hydrolytic water exactly in-line (give the angle)? How would the geometry be different with a magnesium ion (do the respective model)?

Previous response: For clarity, we would prefer to keep our figures the way we prepared them originally, in which all relevant information discussed in the manuscript is shown in a manner that the figure is not overcrowded and is easy to understand. In the case of Figure 2B, we are showing all interactions of the calcium ion within the active site, corresponding to what we are discussing in the manuscript. Adding distance labels on each interaction or displaying amino acids that are not involved in this interaction would make this figure more difficult to read. The information requested by this reviewer is readily available from the PDB coordinates, which will be released upon publication at the latest.

This reviewer is correct that the 6ep0 structure features a calcium ion with a high B factor. However, this calcium is located in a crystal contact and is distinct from the calcium ion located in the active site

which is discussed in the manuscript. To make that point clearer, we have included an omit map of the active site calcium ion from the EffIC-Ca²⁺-ADP structure, which shows the good quality of the electron density map (Supplementary Figure 1C).

The position of the nucleophilic water molecule is modelled in Figure 2B (in grey, already shown in the previous version of the manuscript). We have reworded the description of the heptahedral geometry (lines 142-144) and the discussion on the position of the nucleophilic water (lines 282-285) to make these points clearer .

Response_response. Still would be interested to see whether the calcium with its coordination sphere does not make any other contacts, e.g. to the backbone. Thank you for Fig. S1C and the clarifying text. Heptahedral coordination is new to me, in the provided reference I can't find an explanation. Probably the authors mean 6-fold coordination with one ligand missing and one coordination site interacting with 2 atoms? Italic

New response. We agree that « heptahedral coordination » is unusual, and we have replaced it by « pentagonal bipyramid configuration », which is the term used in Ref 25 (Carafoli et al.). We have also reworded the description of the Ca²⁺ ion coordination to make it clearer (Lines 142-147)

The authors claim that ATPgammaS binds in a competent orientation to wild-type EffIC (Fig. 1E). Are the authors sure that the gamma-phosphate is still present? Do the authors suggest that ATP would bind also competently to EffIC? Why then such a strong boost of activity (to be quantified (!)) upon the E to G mutation? In Fig. 1D, the wtEffIC/ATPgammaS structure is compared with wtNmFic/AMPPNP (NOT ATPgammaS as labeled), 3s6a). For a meaningful comparison the same non-hyphorlyzable ligand should be used and the presence of the gamma-phosphate should be verified.

Previous response : We agree with this reviewer that since the gamma phosphate cannot be located in the electron density, the possibility that it is not present cannot formally be ruled out, similar to previous structures of NmMIC using AMPPNP (e.g. Engel et al., Nature 2010). **However, we were unable to obtain crystals of EffIC in the exact same conditions except that ATPgS was replaced by ADP, which would be in favor of ATPgS being the actual ligand.** Note also that our analysis is focused on the cleavable bond located between the alpha and beta phosphates of the nucleotide, which is fully defined in the electron density and is chemically identical in ATP, AMPPNP and ATPgS.

Response_response. Not fully answered (see in bold italics). The argument in bold is not strong.

It is somewhat amazing that the authors give so little information about their EffIC structures and their oligomeric forms. I checked the 5nv5 structure, which is already available, and it turns out that it is the same tetramer as NmFic (Stanger et al., 2016), stabilised by homologous salt-bridges.

Previous response : We have included the requested analysis of the oligomeric states of EffIC in all crystallographic structures (Figure S2, panels A-C and lines 179-186). Briefly, EffIC forms dimers in all crystal forms, but the arrangement of these dimers differs between crystals. Specifically, depending on the crystal forms, EffIC does not form assemblies larger than the dimers, or forms tetramers related to the NmFIC tetramer, or forms hexamers comprised of trimers of dimers. Thus, the oligomeric state of EffIC cannot be established based on the crystallographic assemblies and it is not always a tetramer. Please note that the crystallographic assemblies of all crystal structures were listed in Supplementary Table 1 (crystallographic summary) in our first submission.

Response-response. Thanks for Fig. S2. So the questions arise what is the Kd of dimer formation and whether the dimer is catalytically competent. The increase in deAMPylation activity with enzyme concentration shown in Fig. S2D is no proof at all that the dimer is active. With increasing

concentration the monomer fraction will monotonically increase always, even when dimers are formed. Not end concentrations are needed, but the dependence of $k_{cat}=v_{init}/E$ on enzyme concentration (is it constant or does it go down?) .

New response. We recall that since the AMPylation efficiency of a related FIC protein (NmFIC) was reported to be adversely affected by dilution, which was correlated with changes in its oligomerization state, we envisioned the possibility that this could be the same for the deAMPylation efficiency of EfFIC. We tested this possibility, as shown in Figure S2D. Our experiments unambiguously show that the deAMPylation efficiency of EfFIC is not adversely affected by increasing its concentration. There is thus no reason to envision that this activity is regulated by a change in oligomerization. An analysis of the oligomeric state of EfFIC at each concentration would thus not yield meaningful insight with respect to the deAMPylation reaction and its regulation by metals, which is the focus of this study.

Also, they discuss the complex regulatory mechanism of the close NmFic homolog reported in this paper only very tersely. This is again amazing, since they show (only qualitative) data about the protein concentration dependence of EfFIC mediated de-AMPylation (Fig. 3a).

Previous response: We do not understand this comment, as we are discussing the regulatory mechanism of NmFIC in the introduction (now lines 69-72, which we have slightly reworded to make it clearer), lines 176-179 in the results, and lines 303-306 in the discussion. Also, our analysis of the protein concentration dependency of EfFIC de-AMPylation (now moved to Supplementary Figure 2D) is not solely qualitative. Although we have not determined the quaternary structure that predominate at each concentration, the quantification clearly shows that deAMPylation increases with EfFIC concentration, hence that deAMPylation is not adversely affected by the toxin concentration.

Response-response. Thanks for the improvement of the description of the NmFic mechanism. For concentration dependence, see my previous comment.